# Role of the visual experience-dependent nascent proteome in neuronal plasticity

**Han-Hsuan Liu[1,2,3], Daniel B McClatchy[4], Lucio Schiapparelli[1,2], Wanhua Shen[1,2,5], John R Yates III[2,4], Hollis T Cline[1,2,3,4]\***

[1]The Dorris Neuroscience Center, The Scripps Research Institute, La Jolla, United States; [2]Department of Neuroscience, The Scripps Research Institute, La Jolla, United States; [3]Kellogg School of Science and Technology, The Scripps Research Institute, La Jolla, United States; [4]Department of Molecular Medicine, The Scripps Research Institute, La Jolla, United States; [5]Zhejiang Key Laboratory of Organ Development and Regeneration, College of Life and Environmental Sciences, Hangzhou Normal University, Hangzhou, China

**Abstract** Experience-dependent synaptic plasticity refines brain circuits during development. To identify novel protein synthesis-dependent mechanisms contributing to experience-dependent plasticity, we conducted a quantitative proteomic screen of the nascent proteome in response to visual experience in *Xenopus* optic tectum using bio-orthogonal metabolic labeling (BONCAT). We identified 83 differentially synthesized candidate plasticity proteins (CPPs). The CPPs form strongly interconnected networks and are annotated to a variety of biological functions, including RNA splicing, protein translation, and chromatin remodeling. Functional analysis of select CPPs revealed the requirement for eukaryotic initiation factor three subunit A (eIF3A), fused in sarcoma (FUS), and ribosomal protein s17 (RPS17) in experience-dependent structural plasticity in tectal neurons and behavioral plasticity in tadpoles. These results demonstrate that the nascent proteome is dynamic in response to visual experience and that *de novo* synthesis of machinery that regulates RNA splicing and protein translation is required for experience-dependent plasticity.

DOI: https://doi.org/10.7554/eLife.33420.001

**\*For correspondence:**
cline@scripps.edu

**Competing interests:** The authors declare that no competing interests exist.

## Introduction

The nervous system remodels by changing circuit connectivity in response to sensory experience. This process, known as synaptic plasticity, is thought to be the cellular basis of learning and memory, as well as experience-dependent development of brain circuitry (*Cline et al., 1996*; *Ho et al., 2011*; *Kandel, 2001*; *Lamprecht and LeDoux, 2004*; *Sutton and Schuman, 2006*). Cells require *de novo* protein synthesis to maintain synaptic plasticity for hours or days, demonstrated using protein synthesis inhibitors or genetic approaches to modify translational efficiency (*Agranoff and Klinger, 1964*; *Chen et al., 2012*; *Flexner et al., 1963*; *Kelleher et al., 2004*; *Sutton and Schuman, 2006*). Both long-term potentiation (LTP) and long-term depression (LTD) of synaptic transmission are blocked by protein synthesis inhibitors (*Krug et al., 1984*; *Linden, 1996*; *Lisman et al., 2002*; *Stanton and Sarvey, 1984*).

Although the requirement for protein synthesis in long-term plasticity is widely recognized, the identities of proteins that are differentially synthesized in response to experience and their functions in neuronal and behavioral plasticity are still largely unknown. Several studies focused on specific candidates based on their known functions in synaptic plasticity, for example alpha calcium/calmodulin-dependent protein kinase type II (αCaMKII), brain-derived neurotrophic factor (BDNF) and cytoplasmic polyadenylation element binding protein (CPEB) (*Miller et al., 2002*; *Schwartz et al., 2011*; *Shen et al., 2014*). These studies demonstrated that regulation of synthesis of individual candidates

is critical for synaptic plasticity but failed to introduce novel candidates. Other studies used label-free synaptic proteomic analysis to identify candidates which changed in abundance in response to activity, but could not determine if the changes resulted from alterations in newly synthesized proteins or preexisting proteins (*Butko et al., 2013*; *Kähne et al., 2016*; *Liao et al., 2007*).

It is challenging to detect changes resulting from differences in *de novo* protein synthesis by comparing the whole proteome between different conditions because the dominant preexisting proteins can mask the changes in newly synthesized proteins (NSPs), which are relatively low-abundance. Bio-orthogonal metabolic labeling (BONCAT) solves this problem by adding a tag to NSPs for enrichment (*Dieterich et al., 2007*). BONCAT allows identification of NSPs following incorporation of non-canonical amino acids, such as azidohomoalanine (AHA), which is incorporated into NSPs in place of endogenous methionine (*Ngo and Tirrell, 2011*). AHA is then tagged with biotin alkyne using click chemistry, followed by direct detection of biotin tags (DiDBiT), a method to increase tandem mass spectroscopic (MS/MS) coverage and sensitivity of detection of biotin-labeled proteins (*Schiapparelli et al., 2014*). We previously combined BONCAT and MS/MS to identify NSPs generated under normal physiological conditions *in vivo* in rat retina (*Schiapparelli et al., 2014*) and in *Xenopus* brain, where we labeled proteins that were newly synthesized over a 24 hr period of development (*Shen et al., 2014*). BONCAT has also been used for quantitative analysis of BDNF-, (RS)−3,5-dihydroxyphenylglycine (DHPG), tetrodotoxin-, or bicuculline-induced proteomic changes *in vitro* (*Bowling et al., 2016*; *Schanzenbächer et al., 2016*; *Zhang et al., 2014*). *In vivo* application of BONCAT as a discovery tool for novel candidate plasticity mechanisms based on quantitative analysis of proteomic changes in response to sensory experience has not been reported.

Visual experience induces plasticity in the developing *Xenopus* visual system from synapses to circuit properties to behavior (*Aizenman et al., 2003*; *Cline, 2016*; *Engert et al., 2002*; *Mu and Poo, 2006*; *Schwartz et al., 2011*; *Shen et al., 2011*; *Sin et al., 2002*). In particular, visual experience induces dendritic arbor plasticity in tectal neurons (*Cline, 2016*) and protein translation-dependent visual avoidance behavioral plasticity (*Shen et al., 2014*). Here we conducted an unbiased quantitative proteomic screen to systematically examine visual experience-induced changes in the nascent proteome in *Xenopus* optic tectum and investigated the role of select candidates in tectal cell structural plasticity and behavioral plasticity. We identified candidate plasticity proteins (CPPs) based on quantitative increases and decreases in the nascent proteome from optic tecta of tadpoles exposed to visual experience compared to controls. CPPs were annotated to several biological functions, including RNA splicing, protein translation, and chromatin remodeling. We showed that synthesis of CPPs, eukaryotic initiation factor three subunit A (eIF3A), fused in sarcoma (FUS), and ribosomal protein s17 (RPS17), are required and work coordinately to facilitate visual experience-dependent structural and behavioral plasticity. These results indicate that synthesis of the machinery that regulates RNA splicing and protein translation is itself tightly controlled in response to visual experience, suggesting that *de novo* synthesis of core cellular machinery is a critical regulatory node for experience-dependent plasticity.

## Results

### Visual experience induces nascent proteome dynamics *in vivo*

To identify NSPs that are differentially synthesized in response to visual experience, we conducted quantitative proteomic analysis using dimethyl labeling in combination with BONCAT with MS/MS analysis, using multidimensional protein identification (MudPIT) (*Figure 1A*). AHA was injected into the midbrain ventricle and tadpoles were exposed to plasticity-inducing visual experience or ambient light. NSPs were tagged with biotin, biotinylated peptides were enriched with DiDBiT and NSPs were identified by detection of biotinylated peptides in MS/MS. The MS/MS spectra were searched against three databases, the Uniprot *Xenopus laevis* database, Xenbase, and PHROG (*Wühr et al., 2014*), and converted to human homologs according to gene symbol. We detected 4833 proteins in the global brain proteome, identified from the unmodified peptides after AHA-biotin enrichment, and 835 AHA-labeled NSPs in the nascent proteome (*Supplementary file 1*). The nascent proteome is comprised of NSPs labeled with AHA over 5 hr in the *Xenopus* optic tectum in animals exposed to visual experience or ambient light.

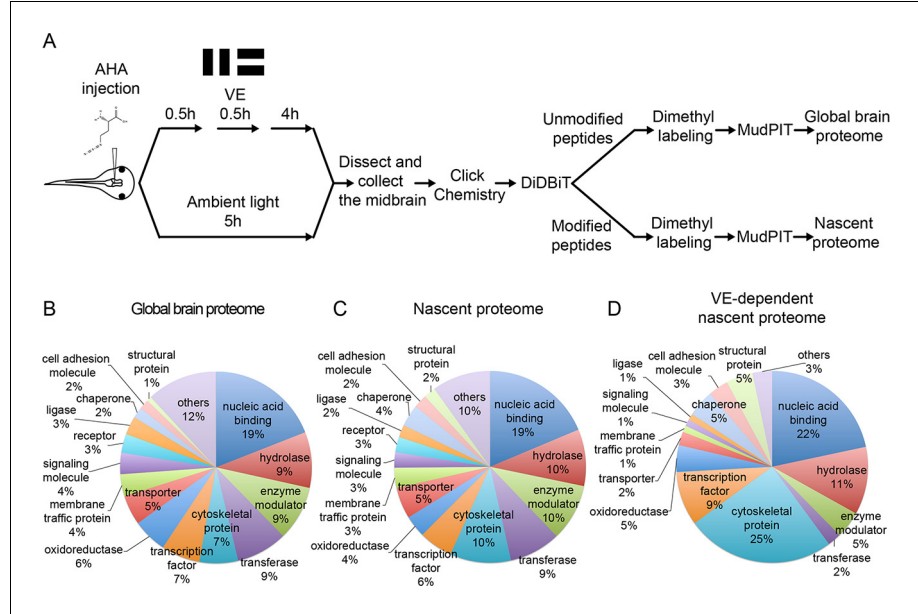

**Figure 1.** Quantitative MS/MS analysis of newly synthesized proteins *in vivo* identifies visual experience-induced dynamics in the nascent proteome. (**A**) Protocol to prepare AHA-labeled samples from animals with or without visual experience (VE) for quantitative proteomic analysis. The midbrain ventricle was injected with AHA before exposure to a moving bar stimulus for 0.5 hr followed by 4 hr in ambient light. Control animals were exposed to ambient light for 5 hr. We dissected midbrains from 1200 to 1500 stage 47/48 tadpoles for each experimental group, yielding about 15 mg protein, in two independent experiments. Newly synthesized proteins (NSPs) from midbrains were tagged with biotin using click chemistry and processed for direct detection of biotin tags (DiDBiT). Samples from control and VE-treated animals were combined after dimethyl labeling for multidimensional protein identification technology (MudPIT) analysis. We identified 83 candidate plasticity proteins (CPPs) in the VE-dependent nascent proteome. (**B–D**) Pie charts of the protein classes in the global brain proteome (**B**), nascent proteome (**C**), and VE-dependent nascent proteome (**D**). Proteins were annotated using PANTHER protein classes.
DOI: https://doi.org/10.7554/eLife.33420.002

The visual experience-dependent nascent proteome, consists of 83 proteins which are a subset of the nascent proteome, in which NSPs have at least a 20% change in synthesis in response to visual experience compared to ambient light (*Table 1*, S2). Comparable percentages of proteins increase (45.8%; 38/83) and decrease (54.2%; 45/83) synthesis in response to visual experience (*Table 1*). The 83 CPPs are annotated to multiple cellular compartments, molecular functions, and biological processes with the PANTHER database (*Supplementary file 3*) (*Mi et al., 2016*). We compared the global brain proteome, the nascent proteome, and the visual experience-dependent nascent proteome based on PANTHER protein classes and found that the visual experience-dependent nascent proteome has a higher percentage of cytoskeletal proteins (25%) than the nascent (10%) and global brain proteomes (7%) (*Figure 1B–D*, *Supplementary file 5*). Furthermore, 54.2% (45/83) of CPPs are localized to synapses according to SynProt classic and PreProt databases from SynProt Portal, a website containing comprehensive synapse-associated proteomics databases, and 30.1% (25/83) are localized to presynaptic sites, including presynaptic vesicles, the cytomatrix and the active zone (*Supplementary file 6*) (*Pielot et al., 2012*). A total of 22.9% (19/83) of our CPPs are autism spectrum disorder (ASD) genes, identified from the Simons Foundation and Autism Research Initiative (SFARI) database (gene.sfari.org), and FMRP targets, identified by CLIP (*Darnell et al., 2011*) (*Table 1*), suggesting that synthesis of these disease genes could be regulated by activity (*Table 1*). By contrast, only 14.5% (702/4833) of the tectum global proteome are ASD genes or FMRP targets (*Supplementary file 1*). Furthermore, the CLIP dataset of FMRP targets is more highly enriched for CPPs (global proteome: 9.4%; CPPs: 19.3%), than the SFARI database (global proteome: 7.4%; CPPs: 7.2%) (*Supplementary file 1*).

**Table 1.** Candidate plasticity proteins (CPPs) include FMRP targets and autism spectrum disorder (ASD) genes.

CPPs were compared to FMRP targets identified by CLIP (**Darnell et al., 2011**) and genes from the Simons Foundation and Autism Research Initiative (SFARI) database (gene.sfari.org) for potential activity-regulated FMRP targets and ASD genes. The final fold changes in AHA labeling is from the average of two independent experiments (**Supplementary file 2**). The 83 CPPs shown here are AHA-labeled newly synthesized proteins (NSPs) that showed consistent increases or decreases in synthesis in response to visual experience with at least 20% change in one of the experiments.

| | Gene names | Human Uniprot ID | Final Fold Change | FMRP targets and ASD genes | | | Gene names | Human Uniprot ID | Final Fold Change | FMRP targets and ASD genes | |
| | | | | FMRP targets | SFARI database | | | | | FMRP targets | SFARI database |
|---|---|---|---|---|---|---|---|---|---|---|---|
| Up in two experiments | capza1 | P52907 | 1.36 | | | Down in two experiments | cct7 | Q99832 | 0.40 | | |
| | capza2 | P47755 | 1.33 | | | | clasp1 | Q7Z460 | 0.74 | V | |
| | cbx1 | P83916 | 1.36 | | | | col18a1 | P39060 | 0.71 | | |
| | cbx3 | Q13185 | 1.36 | | | | ctnnb1 | P35222 | 0.63 | V | V |
| | fus | P35637 | 1.36 | | | | papss1 | O43252 | 0.67 | | |
| | hdlbp | Q00341 | 1.26 | V | | | psmc6 | P62333 | 0.75 | | |
| | rps17 | P08708 | 1.71 | | | | tln1 | Q9Y490 | 0.71 | | |
| | tuba1b | P68363 | 1.95 | | | | atp2a2 | P16615 | 0.66 | V | |
| | lonp1 | P36776 | 1.31 | | | | pmp2 | P02689 | 0.55 | | |
| | sfpq | P23246 | 1.54 | | | | acta1 | P68133 | 0.73 | | |
| | ttpal | Q9BTX7 | 1.42 | | | | atp2a1 | O14983 | 0.66 | | |
| | tuba1a | Q71U36 | 1.95 | | | | kif1a | Q12756 | 0.65 | V | |
| Up in one experiment | cox5a | P20674 | 1.30 | | | Down in one experiment | actb | P60709 | 0.78 | V | |
| | eif3a | Q14152 | 1.16 | | | | aplp2 | Q06481 | 0.74 | | |
| | krt75 | O95678 | 1.17 | | | | cand1 | Q86VP6 | 0.83 | V | |
| | metap2 | P50579 | 1.22 | | | | dnm1l | O00429 | 0.78 | | V |
| | naca | Q13765 | 1.38 | | | | hist1h4a | P62805 | 0.61 | | |
| | nono | Q15233 | 1.15 | | | | hnrnpa1 | P09651 | 0.81 | | |
| | psmd2 | Q13200 | 1.40 | | | | hnrnpc | P07910 | 0.76 | | |
| | rab5a | P20339 | 1.14 | | | | hsp90ab1 | P08238 | 0.79 | V | |
| | rab5c | P51148 | 1.14 | | | | hspa5 | P11021 | 0.68 | | |
| | eif4a1 | P60842 | 1.18 | | | | mdh2 | P40926 | 0.75 | | |
| | hnrnpab | Q99729 | 1.21 | | | | ncl | P19338 | 0.81 | | |
| | pbrm1 | Q86U86 | 1.36 | | | | pcbp3 | P57721 | 0.60 | | |
| | rab5b | P61020 | 1.20 | | | | snw1 | Q13573 | 0.75 | | |
| | skiv2l2 | P42285 | 1.25 | | | | actg1 | P63261 | 0.79 | | |
| | ap2a2 | O94973 | 1.15 | V | | | hn1 | Q9UK76 | 0.82 | | |
| | iws1 | Q96ST2 | 1.25 | | | | kif1b | O60333 | 0.81 | V | |
| | vim | P08670 | 1.19 | | | | pcbp2 | Q15366 | 0.60 | | |
| | nsrp1 | Q9H0G5 | 1.36 | | | | smc1a | Q14683 | 0.80 | | |
| | syp | P08247 | 1.17 | | | | sptan1 | Q13813 | 0.84 | V | |
| | cxxc4 | Q9H2H0 | 1.47 | | | | stip1 | P31948 | 0.74 | | |
| | dpysl3 | Q14195 | 1.19 | | | | trim69 | Q86WT6 | 0.71 | | |
| | fasn | P49327 | 1.23 | V | | | arnt2 | Q9HBZ2 | 0.78 | V | V |
| | kiaa1598 | A0MZ66 | 1.16 | | | | cdh11 | P55287 | 0.80 | | V |

*Table 1 continued on next page*

*Table 1 continued*

| | Gene names | Human Uniprot ID | Final Fold Change | FMRP targets and ASD genes | | | Gene names | Human Uniprot ID | Final Fold Change | FMRP targets and ASD genes | |
| | | | | FMRP targets | SFARI database | | | | | FMRP targets | SFARI database |
|---|---|---|---|---|---|---|---|---|---|---|---|
| Up in one experiment | krt7 | P08729 | 1.17 | | | Down in one experiment | hk1 | P19367 | 0.80 | V | |
| | lgmn | Q99538 | 1.55 | | | | mcm4 | P33991 | 0.76 | | V |
| | sptbn1 | Q01082 | 1.18 | V | | | smarcd2 | Q92925 | 0.84 | | |
| | | | | | | | stmn2 | Q93045 | 0.70 | | |
| | | | | | | | acta2 | P62736 | 0.80 | | |
| | | | | | | | actc1 | P68032 | 0.80 | | |
| | | | | | | | kif1c | O43896 | 0.81 | | |
| | | | | | | | kif5c | O60282 | 0.80 | V | V |
| | | | | | | | smarcd1 | Q96GM5 | 0.84 | | |
| | | | | | | | smarcd3 | Q6STE5 | 0.84 | | |

DOI: https://doi.org/10.7554/eLife.33420.003

DiDBiT identifies NSPs by virtue of the detection of biotin in MS spectra, providing high confidence in calling AHA-labeled proteins. This is particularly valuable for unbiased discovery-based proteomic studies where antibodies used to validate candidates may not be available. We validated several CPPs by western blot, where we compared total and enriched AHA-labeled midbrain protein homogenates from tadpoles exposed to visual experience or ambient light (*Figure 2*). We detected increases in AHA-labeled αCaMKII, as a positive control, and CPPs including FUS, RPS17 and 26S proteasome non-ATPase regulatory subunit 2 (PSMD2). For the non-CPPs, L1CAM and calmodulin, AHA-labeled L1CAM did not change significantly, but AHA-labeled calmodulin decreased significantly in response to visual experience (*Figure 2B*). Although, we rarely detected experience-dependent changes in total protein of individual CPPs by western blot, total αCaMKII levels showed a small but significant decrease with visual experience, and eIF3A increased with visual experience (*Figure 2C*). Similar differences in CPPs were seen after 30 min or 4 hr of visual experience (*Figure 2* and *Figure 2—figure supplement 1*). We could not quantify AHA-labeled eIF3A due to technical issues. Increased synthesis of FUS, RPS17, eIF3A and PSMD2 was detected in the MS/MS experiments (*Supplementary file 2*), indicating that changes in AHA-labeled CPPs detected with western blot corroborate the quantitative proteomic analysis. These results indicate that the nascent proteome is dynamic in response to visual experience and the ability to enrich NSPs from total proteins enables us to observe changes in protein synthesis *in vivo*.

## Bioinformatic analysis indicates that CPPs are enriched in biological processes related to protein translation and RNA splicing

To identify biological processes that may be affected by changes in NSPs, we conducted STRING analysis and pathway enrichment analysis using both human and mouse protein interaction databases (*Szklarczyk et al., 2015*). STRING analysis suggests that CPPs form functional protein interaction networks and within these networks, RNA splicing, protein translation, and chromatin remodeling are the top biological processes predicted to be affected by CPPs (*Figure 3A*). We also analyzed fold changes of CPPs in specific pathways and biological processed identified by STRING (*Figure 3B*). Some CPPs in the RNA splicing and chromatin remodeling modules were synthesized more and others were synthesized less in response to visual experience. Synthesis of CPPs in the protein translation module all increased in response to visual experience (*Figure 3B*). These results indicate that RNA splicing, protein translation, and chromatin remodeling are actively regulated by protein synthesis in response to visual experience, suggesting that *de novo* synthesis of machinery involved in these biological processes could be important for experience-dependent plasticity. We tested this hypothesis in experiments described below.

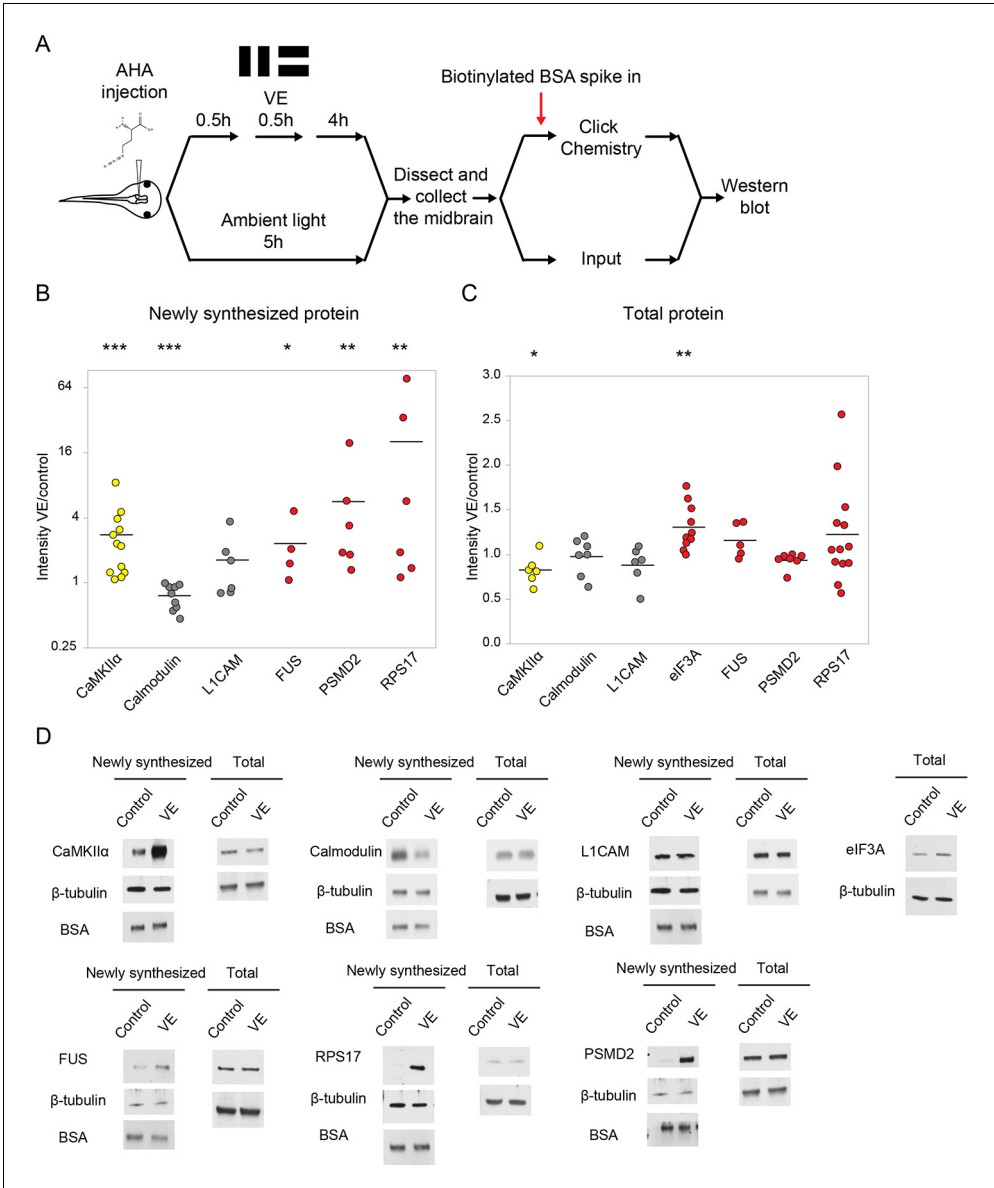

**Figure 2.** Validation of visual experience-dependent changes in CPP synthesis. (**A**) Protocol to prepare AHA-labeled samples. Tissue was processed to tag AHA-labeled proteins from VE-treated and control samples with biotin using click chemistry followed by western blot analysis. (**B, C**) Scatter plots of western blot data of newly synthesized (**B**) or total (**C**) CPPs and non-CPPs. Data are presented as ratios of intensities for paired VE and control samples. αCaMKII (yellow) is a positive control. WBs of CPPs (red) corroborated the proteomic results. For non-CPPs (gray), L1CAM NSPs increased and decreased in western blot and proteomic data, while calmodulin NSPs consistently decreased in western blot data, but increased and decreased in proteomic data (*Supplementary file 2*). (**D**) Representative images of western blots of newly synthesized or total CPPs and non-CPPs. The Y axis of B is plotted in a log scale and the Y axis of C is plotted in a linear scale. *$p < 0.05$, **$p < 0.01$, ***$p < 0.001$, two-tailed Student's t test (**C**) or Mann-Whitney test (**B**) was used to compare between paired samples from control and VE treatments. $n \geq 4$ independent experiments for each CPP. The black bars represent the mean.

DOI: https://doi.org/10.7554/eLife.33420.004

The following source data and figure supplements are available for figure 2:

**Source data 1.** Values of the scatter plots of western blot data presented in *Figure 2B–C*.
DOI: https://doi.org/10.7554/eLife.33420.006

**Figure supplement 1.** Validation of changes in synthesis of eIF3A, FUS, and RPS17 in response to 4 hr of visual experience.

*Figure 2 continued on next page*

*Figure 2 continued*

DOI: https://doi.org/10.7554/eLife.33420.005

**Figure supplement 1—source data 1.** Values of the scatter plots of western blot data presented in *Figure 2—figure supplement 1B-C*.

DOI: https://doi.org/10.7554/eLife.33420.007

## Reduced synthesis of RNA splicing and protein translation machinery blocks visual experience-dependent structural plasticity

To test if synthesis of cellular machinery regulating RNA splicing and protein translation are required for visual experience-dependent structural plasticity, we performed *in vivo* time-lapse imaging of

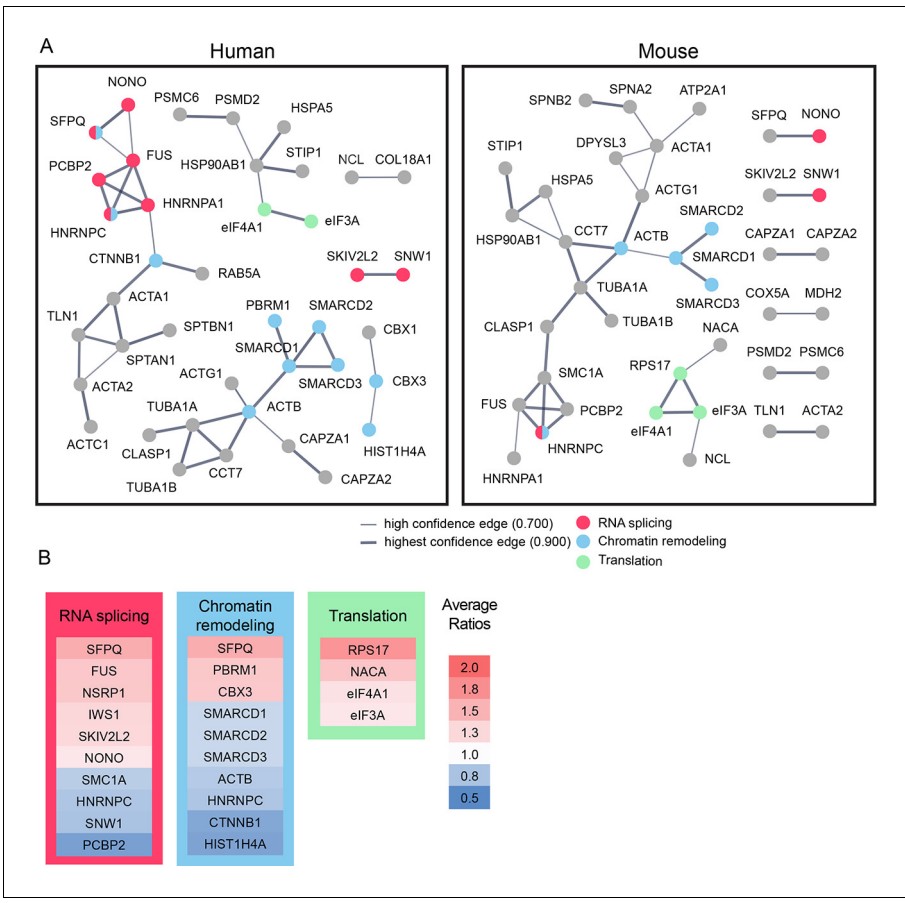

**Figure 3.** Bioinformatic analysis indicates that candidate plasticity proteins are enriched in processes related to protein translation, RNA splicing and chromatin remodeling. (**A**) Functional protein interaction networks of CPPs, shown as dots. Lines connecting CPP nodes represent protein interactions reported in the human (left) or mouse (right) STRING databases, with higher interaction confidence represented by thicker lines. CPPs belonging to the top biological processes in each network are color-coded: RNA splicing (red); chromatin remodeling (blue); and translation (green). (**B**) Fold changes in synthesis of all CPPs pertaining to RNA splicing, translation and chromatin remodeling, not restricted to those within the networks identified by STRING in (**A**), are color coded by average fold increase or decrease compared to control from 0.5 (blue) to 2.0 (red), indicated at the right. See also *Supplementary file 4*, which shows that pathways involved in RNA splicing and chromatin remodeling are statistically enriched for CPPs.

DOI: https://doi.org/10.7554/eLife.33420.008

The following source data is available for figure 3:

**Source data 1.** Values of the fold changes in synthesis of all CPPs presented in *Figure 3B*.

DOI: https://doi.org/10.7554/eLife.33420.009

GFP-expressing tectal neurons in animals exposed to visual experience while synthesis of individual CPPs was blocked using translation-blocking antisense morpholino oligonucleotides (MOs). We selected FUS and non-POU domain-containing octamer-binding protein (NONO) from the RNA splicing network and eIF3A and RPS17 from the protein translation network described in *Figure 3*. FUS, eIF3A, and RPS17 not only form a strong network, but were also validated for their increased synthesis in response to visual experience using western blot (*Figure 2* and *Figure 2—figure supplement 1*), and are therefore strong candidates for further investigation. We also tested NONO in the structural plasticity analysis because increased NONO synthesis in response to visual experience was detected by MS/MS, and because NONO reportedly interacts with FUS in nuclear paraspeckles, which may participate in pre-mRNA splicing (*Neant et al., 2011*; *Shelkovnikova et al., 2014*).

We co-electroporated GFP-expression plasmids and MOs into the optic tectum to block synthesis of FUS, NONO, eIF3A, or RPS17 in response to visual experience. Individual GFP-labeled neurons were imaged on a two-photon microscope before and after 4 hr exposure to dark or visual experience, and their dendritic arbors were reconstructed (*Figure 4A*). The dendritic arbors of control neurons, electroporated with control MO, such as the example shown in *Figure 4A*, grew significantly more, an average of ~30% increase in growth rate (change in TDBL over 4 hr), in response to visual experience compared to dark (*Figures 4C,D* and *5B–D*). Knocking down eIF3A, FUS, and RPS17 blocked the visual experience-dependent increase in growth rate (*Figure 4C', 5B' and B''*), indicating that inhibiting synthesis of eIF3A, FUS, and RPS17 each produced deficits in experience-dependent structural plasticity. Knockdown of NONO did not affect structural plasticity (*Figure 4C''*).

To test whether MOs affect CPP expression, we electroporated tecta with control MO, FUS MO, eIF3A MO, or RPS17 MO and dissected two days later. FUS MO targets the first splice donor site of *fus* mRNA and is predicted to cause inclusion of intron 1 and reduce the amount of both *fus-a* and *fus-b* splice variants by premature termination (*Dichmann and Harland, 2012*). To evaluate FUS MO knockdown, we used real-time PCR (RT–PCR) to assess the level of *fus-a* transcript. We found a significant 22% reduction of *fus-a* transcript compared to control *rps13* transcript (*Figure 4E*). Furthermore, *gria1*, the transcript for AMPA type glutamate receptor subunit 1 (GluA1), which is stabilized by FUS (*Udagawa et al., 2015*) was significantly reduced by 32.2% with FUS knockdown (*Figure 4G*). We validated the knockdown efficiency of eIF3A MO and RPS17 MO with western blot and found that eIF3A MO and RPS17 MO significantly reduced eIF3A and RPS17 protein by 50% and 46%, respectively (*Figure 5E,G*).

We tested whether the deficit in experience-dependent structural plasticity can be rescued by expression of CPPs and found that co-expressing MOs and MO-insensitive eIF3A, FUS, or RPS17 rescue constructs restored the experience-dependent structural plasticity (*Figure 4D'', 5C'' and D''*). The rescue constructs contain the open reading frame of each CPP excluding the 5' UTR MO target sites, followed by t2A and GFP, to identify cells expressing the rescue constructs. Expressing rescue constructs for two days generated 40% and 100% more FUS and eIF3A proteins in tadpole brains (*Figures 4F* and *5F*). RPS17 overexpression makes tectal cells unhealthy, so we transfected HEK cells with the RPS17 rescue construct and found that RPS17 immunolabeling intensity was significantly stronger in GFP positive cells compared to GFP negative cells, indicating that cells with the rescue construct express more RPS17 (*Figure 5H*). Overexpressing FUS and eIF3A does not appear to interfere with structural plasticity (*Figure 4—figure supplement 1*). For the RPS17 overexpressing tectal neurons that we could reconstruct, their dendritic arbors failed to show structural plasticity (*Figure 4—figure supplement 1*). Therefore, the rescue of experience dependent plasticity is likely achieved by restoring functional levels of FUS, eIF3a and RPS17 in the presence of MOs. Together, these data indicate that experience-dependent increased synthesis of FUS, eIF3A, and RPS17 is required for experience-dependent structural plasticity.

Blocking both protein translation and RNA splicing has profound effects on visual experience-dependent structural and behavioral plasticity eIF3A and RPS17 are part of the 43S pre-initiation complex and may function coordinately to regulate protein translation. We tested the effect of double knockdown of eIF3A and RPS17 on visual experience-dependent structural plasticity and found that knocking down both eIF3A and RPS17 blocked the experience-dependent dendritic arbor growth rate relative to growth rate in the dark (*Figure 6A,B,B'*). Direct comparison of the effects of single or combination MO conditions on dendritic arbor growth rates over 4 hr in dark indicates that neurons in tecta treated with FUS MO, eIF3A MO or RPS17 MO individually or with both eIF3A MO plus RPS17 MO together have no differences in dendritic arbor growth over 4 hr in dark (*Figure 6E*).

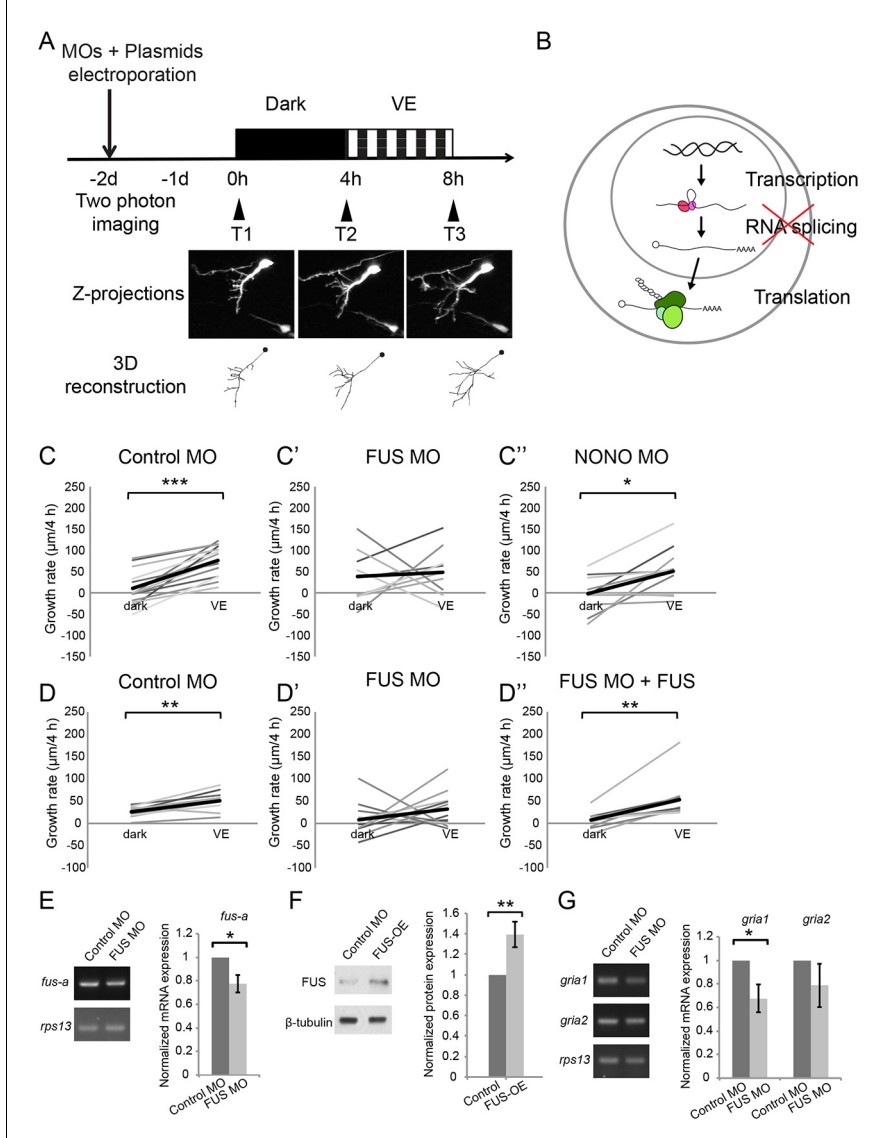

**Figure 4.** Newly synthesized FUS is required for visual experience-dependent structural plasticity. (**A**) Protocol to test the effect of MO-mediated CPP knockdown on VE-dependent structural plasticity. Tecta were co-electroporated with MOs and GFP-expression plasmid 2 days before imaging. GFP-expressing tectal neurons were imaged *in vivo* before and after 4 hr in dark followed by 4 hr of VE. Images of a control neuron are shown. Dendritic arbors of individual neurons were reconstructed and total dendritic branch length (TDBL) was compared across imaging time-points. (**B**) A schematic of different regulatory steps of gene and protein expression, including nuclear transcription, RNA splicing, and cytosolic translation. (**C–C''**) Plots of VE-dependent growth rates (changes in TDBL over 4 hr) in dark and VE in tecta electroporated with control MO (**C**), FUS MO (**C'**), or NONO MO (**C''**). Gray lines connect data points for individual neurons and black lines are average growth rates in dark and VE. Neurons treated with control MO increase growth rate with VE compared to dark. FUS MO blocked the normal increase in growth rate in response to VE. VE-dependent structural plasticity was unaffected by NONO MO. Control MO: n = 14 cells; FUS MO: n = 9 cells; NONO MO: n = 10 cells. (**D–D''**) The impaired experience-dependent structural plasticity seen with FUS knockdown (**D'**) was rescued by expression of exogenous FUS (**D''**). Control MO: n = 9 cells; FUS MO: n = 11 cells; FUS MO + FUS: n = 10 cells. (**E, F**) Validation of FUS knockdown and overexpression (OE). Normalized mRNA or protein expression of *fus*/FUS in tecta electroporated with FUS MO (**E**) or FUS expression construct (**F**), compared to controls. (**E**) Left: Representative gels of *fus-a* and control *rps13* transcripts from tecta electroporated with control or FUS MO. Right: *fus-a* expression normalized to *rps13* from tecta treated with control or FUS MO. FUS knockdown significantly reduced *fus-a* (0.78 ± 0.08, p=0.0302), n = 4 independent experiments. (**F**) Left. Representative blots of FUS and β-tubulin expression from tecta electroporated with control or FUS expression construct. Right: FUS expression normalized to β-tubulin from tecta

*Figure 4 continued on next page*

*Figure 4 continued*

treated with control or FUS expression constructs. FUS expression construct significantly increased FUS protein. FUS-OE: 1.4 ± 0.13, p=0.0172; n = 5 independent experiments. (G) Left: Representative gels of *gria1* and *gria2* and control *rps13* transcripts from tecta electroporated with control MO or FUS MO. Right: plots of *gria1* and *gria2* expression normalized to *rps13* from tecta treated with control or FUS MO. Fus MO significantly decreased *gria1* (0.68 ± 0.12, p=0.0365) but not *gria2* (0.79 ± 0.18, p=0.1642); n = 4 independent experiments. *p<0.05, **p<0.01, ***p<0.001, two-tailed paired Student's t test for comparisons between two matched pairs (**C–D**) and one-tailed Student's t test for comparisons of two independent groups (**E–G**). Error bars represent ±SEM (**E–G**).

DOI: https://doi.org/10.7554/eLife.33420.010

The following source data and figure supplements are available for figure 4:

**Source data 1.** Values of VE-dependent changes in tectal neuron growth rate over 4 hr in dark and VE presented in *Figure 4C–C''* and *Figure 4D–D''*.

DOI: https://doi.org/10.7554/eLife.33420.012

**Figure supplement 1.** Newly synthesized FUS is required for visual experience-dependent structural plasticity.

DOI: https://doi.org/10.7554/eLife.33420.011

**Figure supplement 1—source data 1.** Values of the plots of VE-dependent changes in tectal neuron growth rate over 4 hr in dark and VE presented in *Figure 4—figure supplement 1A–D*.

DOI: https://doi.org/10.7554/eLife.33420.013

By contrast, neurons treated with eIF3A MO plus RPS17 MO had a significantly lower visual experience-dependent dendritic growth rate compared to controls (*Figure 6F*). These data show that double knockdown of eIF3A and RPS17 exhibits stronger deficits in experience-dependent structural plasticity compared with single knockdown of eIF3A or RPS17.

To assess whether knockdown of eIF3A and RPS17, individually or in the double knock down condition, affects overall protein synthesis, we used fluorescent non-canonical amino acid tagging (FUN-CAT) to visualize *in vivo* AHA-labeled NSPs in the optic tectum, as previously described (*Liu and Cline, 2016*). AHA labeling increased significantly in the neuronal cell body layer and the neuropil of animals treated with eIF3A MO compared to animals treated with control MO (*Figure 6—figure supplement 1*). Double knockdown of eIF3A and RPS17 increased AHA labeling in the neuropil, but not in the cell body layer. These results suggest that the effects of RPS17 and RPS17 knockdown in visual experience-dependent structural plasticity are not due to large-scale decreases in protein synthesis.

Animals are reared in a 12 hr light/dark cycle before experiments begin. We tested the effects of CPP knockdown on basal levels of dendritic arbor growth during development by comparing dendritic arbor structure (TDBL) at the first imaging timepoint before the visual experience protocol, and found dendritic arbors in tecta treated with both eIF3A MO and RPS17 MO for 2 days were significantly less complex than controls (*Figure 6G*).

We next tested whether simultaneously interfering with both eIF3A- and RPS17-mediated protein translation and FUS-mediated RNA splicing would have more severe deficits compared to knockdown of individual candidates or double knockdown of candidates involved in the same biological function. We knocked down FUS together with eIF3A and RPS17 by electroporating a mixture of FUS, eIF3A and RPS17 MOs into the tectum, and found that the visual experience-induced structural plasticity was blocked (*Figure 6C–D'*). Average growth rates over 4 hr in dark with FUS, eIF3A and RPS17 MOs were not significantly different than control or other knockdown conditions, but this treatment blocked the visual experience-dependent structural plasticity (*Figure 6E,F*). Moreover, treatment with FUS, eIF3A and RPS17 MOs significantly reduced developmental dendritic arbor growth, as seen with double eIF3A and RPS17 knockdown (*Figure 6G*). Finally, analysis of the proportion of neurons with visual experience-dependent dendritic arbor growth in each experimental condition indicates that only 29% of neurons in tecta with triple knockdown of eIF3A, RPS17, and FUS show experience-dependent dendritic arbor structural plasticity, the lowest percentage of VE-responsive cells. By contrast, 97% of control neurons showed experience-dependent plasticity, compared to 59–70% of neurons with single knockdown and 55% of neurons with double knockdown of eIF3A and RPS17 (*Figure 6H*). Chi Square analysis of independence with Bonferroni correction indicates that the proportions of neurons that respond to VE in control morpholino and triple knockdown conditions are significantly different (*Figure 6H*). Note that neurons treated with different

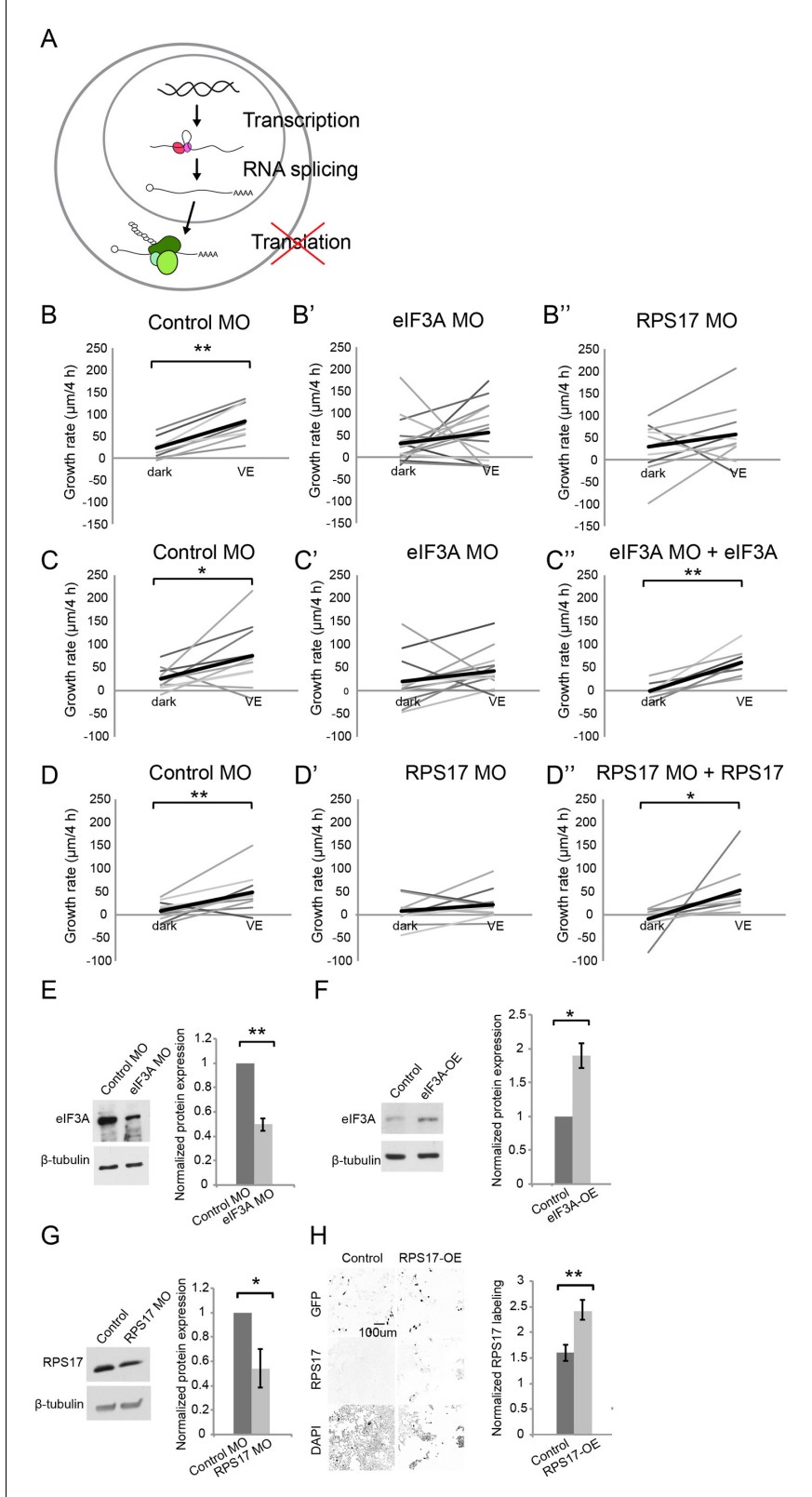

**Figure 5.** Newly synthesized eIF3A and RPS17 are required for visual experience-dependent structural plasticity. (**A**) Schematic of different steps of regulation in gene and protein expression. (**B–D''**) VE-dependent changes in tectal neuron dendritic arbor growth rate over 4 hr in dark and VE in tecta electroporated with control MO (**B, C, D**), designated CPP MOs (**B', B'', C', D'**), or CPP MOs and rescue constructs (**C'', D''**). Gray lines connect data

*Figure 5 continued on next page*

*Figure 5 continued*

points for individual neurons and black lines are average changes in TDBL in dark and VE. (**B–B"**) VE-dependent changes in growth rate in tecta treated with control MO (B, eIF3A MO (**B'**), or RPS17 MO (**B"**). Control MO: n = 22 cells; eIF3A MO: n = 17 cells; RPS17 MO: n = 10 cells. Both eIF3A MO and RPS17 MO blocked the VE-dependent increase in dendritic arbor growth rate observed in controls. (**C–C"**) Co-expression of eIF3A MO and exogenous eIF3A rescued the deficit in VE-induced structural plasticity seen with eIF3A knockdown. Control MO: n = 10 cells; eIF3A MO: n = 14 cells; eIF3A MO + eIF3A: n = 7 cells. (**D–D"**) Co-expression of RPS17 MO and exogenous RPS17 rescued the deficit in VE-induced dendritic structural plasticity seen with RPS17 knockdown. Control MO: n = 9 cells; RPS17 MO: n = 10 cells; RPS17 MO + RPS17: n = 9 cells. (**E–H**) Validation of eIF3A and RPS17 knockdown and OE. (**E, F**) Left: representative WB of eIF3A and β-tubulin from tecta electroporated with eIF3A MO (**E**) or MO-insensitive eIF3A expression construct (**F**) compared to controls. Right: eIF3A MO significantly decreased synthesis of eIF3A protein (eIF3A MO: 0.5 ± 0.05, p=0.0003; n = 5 independent experiments) and the eIF3A expression construct generated significantly more eIF3A protein (eIF3A-OE: 1.89 ± 0.18, p=0.0198; n = 3 independent experiments). (**G**) Left: WB of RPS17 and β-tubulin from tecta electroporated with control or RPS17 MO. Right: Normalized RPS17 expression levels in tecta electroporated with control or RPS17 MO. RPS17 MO significantly reduced synthesis of RPS17 protein (RPS17 MO: 0.54 ± 0.16, p=0.0309; n = 4 independent experiments). (**H**) Left: Images of GFP (top), RPS17 (middle) expression and DAPI (bottom) labeling in HEK cells expressing GFP alone (Control, left) or GFP and RPS17 (right). Right: RPS17 expression in GFP$^+$ ROI, normalized to RPS17 expression in GFP$^-$ ROI. The RPS17 expression construct increased RPS17 immunolabeling. Control: 1.6 ± 0.16; RPS17-OE: 2.43 ± 0.19; p=0.0029; n = 7 different fields imaged from two independent experiments for each experimental condition. *p<0.05, **p<0.01, two-tailed paired Student's t test were used to compare between two matched pairs (**B–D**) and one-tailed Student's t test for comparisons of two independent groups (**E–H**). Error bars represent ±SEM (**E–H**).

DOI: https://doi.org/10.7554/eLife.33420.014

The following source data is available for figure 5:

**Source data 1.** Values of VE-dependent changes in tectal neuron growth rate over 4 hr in dark and VE presented in *Figure 5B–B"*, *Figure 5C–C"*, and *Figure 5D–D"*.

DOI: https://doi.org/10.7554/eLife.33420.015

combinations of MOs all grew comparably to controls over 4 hr in dark (*Figure 6E*) and the deficiency in experience-dependent structural plasticity was only observed in tecta with double or triple knockdown (*Figure 6F*). Taken together, these data indicate that simultaneously blocking synthesis of multiple CPPs that are each necessary for structural plasticity has a more profound effect than single or double knockdown, and that simultaneously interfering with distinct biological functions exacerbates deficits in experience-dependent structural plasticity.

To determine the functional consequences of interfering with both eIF3A- and RPS17-mediated protein translation and FUS-mediated RNA splicing in the optic tectum on visual experience-dependent plasticity, we examined visual avoidance behavioral plasticity in triple knockdown animals. Visual avoidance behavior is an innate behavior where tadpoles change swimming direction to avoid an approaching object (*Dong et al., 2009*). The behavior is quantified as an avoidance index (AI), the ratio of avoidance responses out of 10 encounters with an approaching visual stimulus (*Shen et al., 2011*). AI improves after tadpoles are exposed to visual experience and this behavioral plasticity requires protein synthesis (*Shen et al., 2014*).

Baseline avoidance behavior was tested two days after optic tecta were electroporated with control MO or eIF3A MO + RPS17 MO + FUS MO, then animals were exposed to 4 hr visual experience and tested for visual avoidance behavior 1 hr and 20 hr later (*Figure 7A*). Electroporation delivers morpholinos to neurons throughout the optic tectum (*Bestman and Cline, 2014*), allowing investigation of effects of knockdown on circuit properties (*Shen et al., 2014*). We found no significant difference in baseline avoidance behavior before animals were exposed to visual experience (AI: control MO = 0.28 ± 0.02; eIF3A MO + RPS17 MO + FUS MO = 0.27 ± 0.03). Control animals showed significantly improved AI scores when tested 1 hr after visual experience and AI scores remained elevated the following day (*Figure 7B*). By contrast, visual experience did not improve AI scores in triple knockdown animals when behavior was tested either 1 hr or 1 day after the visual experience (*Figure 7C*). These data indicate that simultaneous knockdown of CPPs involved in translation and RNA splicing in the optic tectum blocks visual experience-dependent behavioral plasticity.

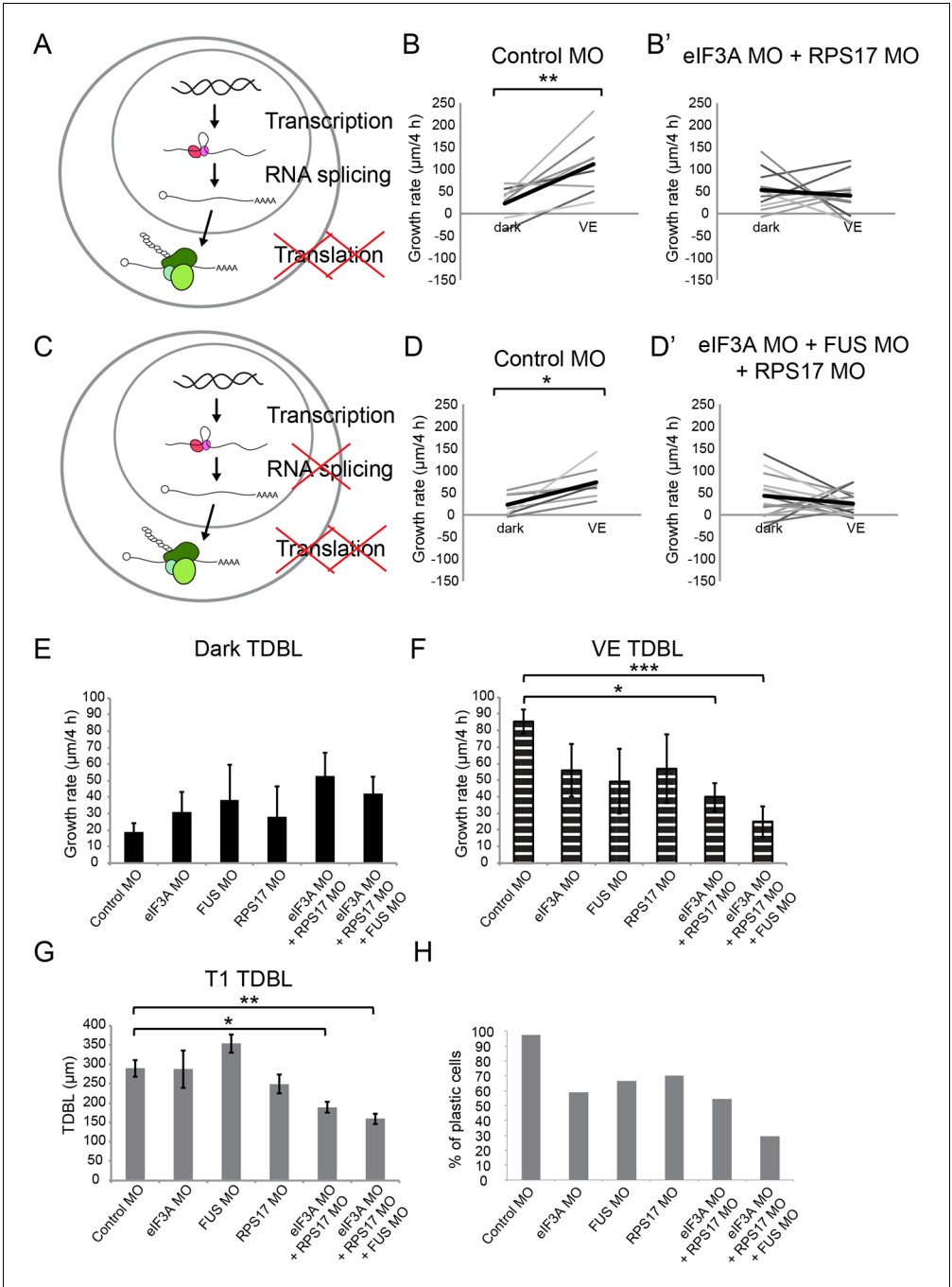

**Figure 6.** Knockdown of EIF3a, RPS17, and FUS blocks visual experience-dependent structural plasticity. (**A, C**) Schematics of different steps of regulation in gene and protein expression. We tested the effect of manipulating translation alone (**A**) or both translation and RNA splicing (**C**). (**B–B', D–D'**) VE-dependent changes in growth rate over 4 hr in dark and VE in tecta treated with control MO (**B, D**) or designated CPP MO mixtures (**B', D'**). Gray lines connect data points for individual neurons and black lines are average changes in growth rate in dark and VE. (**B, B'**) VE-dependent changes in growth rate in neurons from tecta electroporated with control MO (**B**) or eIF3A and RPS17 MO (**B'**). Control MO: n = 9 cells; eIF3A MO + RPS17 MO: n = 11 cells. Knocking down both eIF3A and RPS17 blocked the VE-dependent increase in dendritic arbor growth. (**D, D'**) VE-dependent changes in growth rate of neurons in tecta electroporated with control MO (**D**) or a mixture of eIF3A MO, RPS17 MO, and FUS MO (**D'**). Control MO: n = 7 cells; eIF3A MO + RPS17 MO + FUS MO: n = 17 cells. Combined knockdown of eIF3A, RPS17, and FUS blocked the VE-dependent increase in dendritic arbor growth. (**E, F**) Dendritic arbor growth rate over 4 hr in dark (**E**) and VE (**F**) in tecta electroporated with control MO or designated CPP MOs. Dendritic growth

*Figure 6 continued on next page*

*Figure 6 continued*

rates in dark (**E**) are similar to individual MO knockdown but growth rates over 4 hr in VE (**F**) were significantly decreased in tecta electroporated with eIF3A MO + RPS17 MO or eIF3A MO + RPS17 MO + FUS MO. The variances in growth rates in the dark were not significantly different between groups (O'Brien, Brown-Forsythe, Levene, and Bartlett test). (**G**) TDBL at T1 reflects developmental dendritic arbor growth. Combined knockdown of eIF3A, RPS17, and FUS or eIF3A and RPS17 significantly reduced TDBL at T1. (**E–G**) Control MO: n = 38 cells; eIF3A MO: n = 17 cells; FUS MO: n = 9 cells; RPS17 MO: n = 10 cells; eIF3A MO + RPS17 MO: n = 11 cells; eIF3A MO + RPS17 MO + FUS MO: n = 17 cells. (**H**) Percentage of cells that increased dendritic arbor growth rate in response to VE. Control MO: 97%; eIF3A MO: 59%; FUS MO: 67%; RPS17 MO: 70%; eIF3A MO + RPS17 MO: 55%; eIF3A MO + RPS17 MO + FUS MO: 29%. Triple knockdown of eIF3A, RPS17, and FUS resulted in the lowest percentage of VE-responsive cells. Control morpholino and triple knockdown conditions have significantly different proportions of cells that respond to VE compared to the rest of the groups. *p<0.05, **p<0.01, two-tailed paired Student's t test were used to compare between two matched pairs (**B, D**) or Steel-Dwass test with control for nonparametric multiple comparisons (**E–G**). The Chi-Square test for independence with a Bonferroni correction was used to compare distributions of each group with rest of the groups (**H**). Error bars represent ±SEM (**E–G**).
DOI: https://doi.org/10.7554/eLife.33420.016

The following source data and figure supplements are available for figure 6:

**Source data 1.** Values of VE-dependent changes in tectal neuron growth rate over 4 hr in dark and VE presented in *Figure 6B–B'* and *Figure 6D–D'*.
DOI: https://doi.org/10.7554/eLife.33420.018

**Figure supplement 1.** eIF3A and RPS17 regulate protein synthesis in the tadpole tectum.
DOI: https://doi.org/10.7554/eLife.33420.017

**Figure supplement 1—source data 1.** Raw values of normalized AHA labeling in the neuronal cell body layer and neuropil in animals treated with control MO, eIF3A MO, RPS17 MO, or both eIF3A and RPS17 MO presented in *Figure 6—figure supplement 1C*.
DOI: https://doi.org/10.7554/eLife.33420.019

## Discussion

Protein synthesis is required for long lasting neuronal and circuit plasticity but the identities of differentially synthesized proteins required for plasticity are still largely unknown. Different strategies to identify candidate mechanisms regulating plasticity *in vivo* are needed to understand mechanisms regulating experience dependent brain plasticity. In this study, we performed quantitative proteomic analysis in response to a protein translation-dependent plasticity-inducing visual experience protocol in *Xenopus laevis* tadpoles using BONCAT labeling and DiDBiT. We identified 83 CPPs, many of which had not been reported to affect neuronal plasticity and therefore represent novel CPPs. CPP network analysis identified core biological processes, such as RNA splicing and protein translation, that are actively regulated at the level of protein synthesis by sensory experience, which hasn't been reported previously. Our data demonstrate that *de novo* synthesis of components of RNA splicing and protein translation machinery is required for structural and behavioral plasticity, suggesting novel mechanisms for regulation of experience-dependent plasticity. Considering that for any CPP, NSPs may be a fraction of the total protein, these data suggest the experience-dependent NSPs may play a privileged role in regulating plasticity.

Prior studies used strong induction protocols, such as kainic acid injection or electroconvulsive stimuli, to identify activity-induced genes. The candidate plasticity genes were subsequently tested under more physiological conditions, using a variety of assays for molecular, synaptic, structural and behavioral plasticity *in vivo* or *in vitro* (*Leslie and Nedivi, 2011*). Our unbiased proteomic screen was designed to identify candidate plasticity proteins based on *in vivo* experience-induced differences in NSP levels in response to relatively brief exposure to a naturalistic visual experience protocol previously shown to induce protein synthesis dependent behavioral plasticity (*Shen et al., 2014*). We then validated and tested CPP function in several visual experience protocols known to induce structural and behavioral plasticity (*Shen et al., 2014*; *Sin et al., 2002*), and further demonstrated that individual CPP knockdown did not affect basal neuronal development of dendritic arbors. Identification and validation of CPPs using this strategy strengthen the conclusion that these CPPs function in experience-dependent plasticity mechanisms. Our *in vivo* screen would not be expected to detect NSPs that require strong induction conditions, that are rapidly synthesized and degraded or

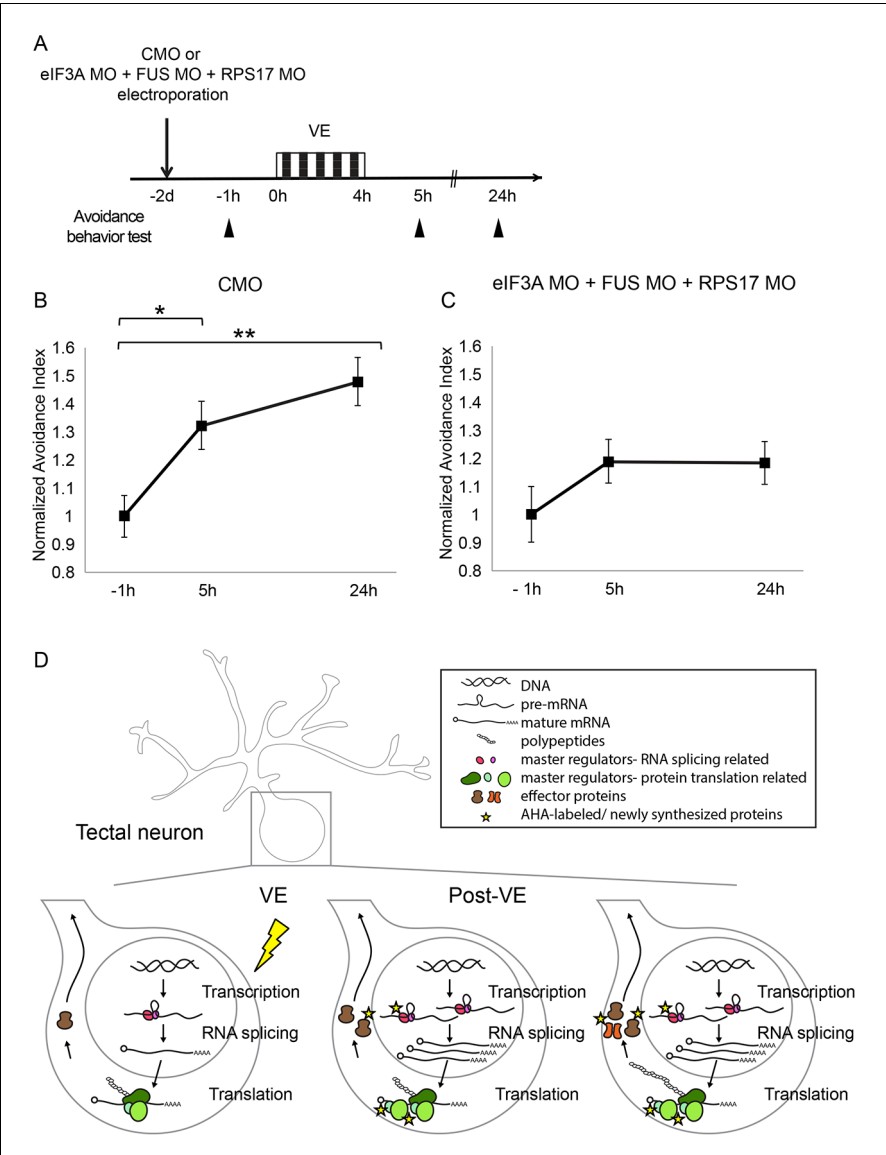

**Figure 7.** Knockdown of eIF3A, RPS17, and FUS blocks visual experience-dependent behavioral plasticity. (**A**) Protocol to evaluate the effect of eIF3A, FUS, and RPS17 knockdown on visual avoidance behavior. Tecta were electroporated with control MO or eIF3A MO + RPS17 MO + FUS MO and 2 days later were assayed for baseline visual avoidance behavior at −1 hr, exposed to VE for 4 hr and assayed for visual avoidance behavior 5 hr and 24 hr after the onset of VE. (**B**) VE-induced behavioral plasticity in control animals was detected at 5 hr and maintained at 24 hr. (**C**) Triple knockdown blocked behavioral plasticity. AI values were normalized to baseline at −1 hr (**B–C**). (**B–C**) CMO: AI = 1 ± 0.07, 1.32 ± 0.09, or 1.48 ± 0.09; n = 21, 27, or 31 animals for tests at −1 hr, 5 hr, or 24 hr; eIF3A MO + RPS17 MO + FUS MO: AI = 1 ± 0.10, 1.19 ± 0.08, or 1.18 ± 0.08; n = 22, 33, or 33 animals for tests at −1 hr, 5 hr, or 24 hr. *=$p < 0.05$, **=$p < 0.01$, the Steel-Dwass test was used for nonparametric multiple comparisons with control. Error bars represent ±SEM. (**D**) Schematic of visual experience-induced dynamics in protein synthesis machinery. By combining BONCAT and DiDBiT, we identified CPPs including master regulators that control gene expression or protein translation, such as eIF3A, FUS, and RPS17, and effector proteins that maintain plasticity in response to VE, such as cytoskeletal proteins. Functional analysis of select CPPs suggests that *de novo* synthesis of master regulators is required for experience-dependent plasticity. Our data suggest that the experience-dependent changes in the nascent proteome result from a combination of direct changes in synthesis of master regulators and effector proteins, and secondary effects downstream of differential synthesis of master regulators.

DOI: https://doi.org/10.7554/eLife.33420.020

The following source data is available for figure 7:

*Figure 7 continued on next page*

*Figure 7 continued*

**Source data 1.** Quantification of normalized AI values data presented in *Figure 7B–C*.
DOI: https://doi.org/10.7554/eLife.33420.021

NSPs that are synthesized at delayed time-points after plasticity-induction. Indeed, our CPP dataset does not overlap with previously reported immediate early genes or candidate plasticity genes induced by strong induction protocols (*Leslie and Nedivi, 2011*; *West and Greenberg, 2011*), however 7/83 (8.4%) of CPPs were recently identified as visual stimulus-responsive genes in mouse visual cortex (*Hrvatin et al., 2018*) and 8/83 (9.6%) of CPPs from our dataset were identified as changing in mouse hippocampus in response to 21 days of environmental enrichment (*Alvarez-Castelao et al., 2017*). Further studies using a variety of experimental protocols will be required to generate more comprehensive datasets of activity-regulated changes in NSPs and identification of CPPs.

## Proteomic analysis of NSP dynamics

Analysis of the visual experience-dependent nascent proteome revealed several interesting facts. First, we observed not only increases, but also decreases in NSPs in response to visual experience. Previous studies showed that inhibiting protein synthesis globally blocks plasticity, however, our data indicating that the synthesis of some CPPs was reduced during induction of plasticity suggest that experience-dependent changes in protein synthesis related to neuronal plasticity may be more fine-tuned and complex than previously thought. The observation that comparable numbers of CPPs increase and decrease synthesis in response to visual experience further suggests that a general increase in basal translation does not account for visual experience-dependent structural plasticity or behavioral plasticity, however the valence of responses of NSPs to visual experience could reflect corresponding increases and decreases in activity-regulated gene expression. Given the extensive regulation of post-transcriptional and translational mechanisms in the nervous system, for instance by RNA binding proteins (*Darnell et al., 2011*; *Klann and Dever, 2004*; *Udagawa et al., 2015*), direct comparisons between activity-regulated genes and NSPs in response to the same stimulation conditions should reveal interesting spatial and temporal complexities of the relation between activity-induced transcription and translation. The mRNA binding protein, FMRP, is a good example of this complexity. FMRP is thought to inhibit translation of its target mRNAs downstream of activity-dependent phosphorylation (*Bartley et al., 2016*; *Ceman et al., 2003*). About 20% of the CPPs we identified are FMRP targets (*Darnell et al., 2011*), the majority of which decreased their synthesis in *Xenopus* optic tectum with visual experience. Decreased FMRP expression, as occurs in Fragile X Syndrome, is thought to impair behavioral plasticity by increasing synthesis of FMRP targets. We previously reported that FMRP knockdown blocks maintenance but not induction of experience-dependent behavioral plasticity, suggesting that in healthy brains FMRP limits translation of proteins that interfere with the maintenance of plasticity (*Liu and Cline, 2016*). Our present analysis of experience-dependent NSP dynamics indicates that neurons maintain a delicate balance in protein synthesis by taking advantage of regulatory mechanisms specialized to increase or decrease distinct subsets of proteins.

Second, we found that the percentage of cytoskeletal proteins in the experience-dependent NSPs is greater than the nascent or global proteomes. It is widely recognized that cytoskeletal dynamics underlie experience-dependent structural plasticity in neurons (*Benito and Barco, 2015*; *Leslie and Nedivi, 2011*; *Van Aelst and Cline, 2004*). Although studies have identified specific cytoskeletal proteins and their regulators that undergo dynamic rearrangements in response to activity (*Lamprecht and LeDoux, 2004*; *Sin et al., 2002*; *Tada and Sheng, 2006*), whether experience-dependent *de novo* synthesis of cytoskeletal proteins is required for neuronal or behavioral plasticity is still unclear. Our unbiased search of experience-regulated NSPs indicates that cytoskeletal proteins are a predominant category that is differentially synthesized under plasticity-inducing conditions. Previous transcriptional screens identified cytoskeletal mRNAs (*Cajigas et al., 2012*; *Moccia et al., 2003*). Our findings provide direct evidence that translational regulation of cytoskeletal proteins is an important mechanism for experience-dependent control of neuronal structure

Third, about 20% of our CPPs have been identified as ASD genes or FMRP targets (*Darnell et al., 2011*). The fact that their translation is visual experience-dependent suggests that the neuronal phenotypes observed in FXS or ASD patients caused by mutation or malfunction of these genes and proteins could arise from mis-regulation of their synthesis in response to sensory experience or plasticity-inducing conditions. Furthermore, about 65% of the CPPs are annotated as synaptic proteins, according to the SynProt classic and PreProt databases, indicating that diverse protein constituents of synapses are dynamically regulated by translational mechanisms. Label-free synaptic proteomic analysis identified changes in protein abundance in response to activity *in vitro* and *in vivo* (*Butko et al., 2013*; *Kähne et al., 2016*; *Liao et al., 2007*). Although these studies suggested novel candidates affecting plasticity, they did not distinguish between preexisting proteins and NSPs.

Fourth, STRING analysis identified functional protein interaction networks of CPPs and biological processes that may be regulated by visual experience-dependent changes in protein synthesis. The interaction networks identified with human and mouse databases are slightly different because different topics are studied in mice and human tissue. For example, the human database has more references for RNA splicing than the mouse database. For protein translation, the situation is reversed. The mouse database is curated to include an interaction between eIF3A and RPS17, which is missing in the human database, based on studies in mice examining the structure of the 43S preinitiation complex (*Hashem et al., 2013*; *Jackson et al., 2010*). Therefore, combining data from two databases allows a more complete analysis of the functions/pathways enriched in CPPs.

Finally, when we compared the nascent proteome comprised of NSPs labeled with AHA over 5 hr in the *Xenopus* optic tectum to our prior dataset of NSPs labeled over 24 hr of normal development in the entire tadpole brain (*Shen et al., 2014*), we find that only 104 proteins overlap between two datasets (*Supplementary file 8*). The differences in the datasets likely reflect differences in the AHA labeling periods, the brain regions and the visual experience conditions between two experiments and suggest that NSPs required to generate basic components of the entire developing nervous system differ from those associated with development and plasticity of the optic tectal circuit. In addition, we note that targeted search for NSPs using BONCAT combined with western blots can be used to validate CPPs that are challenging to detect in unbiased proteomic screens, based on high sensitivity of these methods (*Shen et al., 2014*). Further studies using a variety of stimulus conditions and time points for NSP analysis will be required to generate a comprehensive understanding of proteomic dynamics contributing to neuronal plasticity during development, learning and aging.

To further examine the roles of CPPs in structural plasticity, we selected four candidates, FUS, NONO, eIF3A, and RPS17, from networks identified by the STRING analysis. We measured changes in dendritic arbor elaboration, which is highly correlated to the number of synaptic inputs (*Li et al., 2011*), as well as the complexity and function of brain circuits (*Haas et al., 2006*; *Sin et al., 2002*). Knocking down FUS, eIF3A, and RPS17 significantly decreased dendritic arbor plasticity in response to visual experience. These deficits were rescued when we co-expressed MO-insensitive forms of the transcripts with the MOs, indicating that the deficits resulted from decreased synthesis of the individual candidates. In these experiments, we electroporated tecta with translation blocking morpholinos and evaluated visual experience-dependent structural plasticity two days later. Our proteomic analysis indicates that synthesis of these CPPs specifically increases in response to visual experience. Furthermore, our analysis of dendritic arbor structure indicates that individual morpholino treatments did not interfere with basal arbor development before animals were exposed to the visual experience protocol. Together, these results indicate that blocking the visual experience-induced *de novo* synthesis of the CPPs interferes with structural plasticity, however, depending on basal levels of CPP proteostasis, it is possible that blocking CPP translation over 1–2 days before the visual experience protocol could contribute to the impaired experience-dependent structural plasticity. In previous studies, we found that delivery of CPEB morpholinos immediately before the visual experience protocol, blocked tectal cell structural and functional plasticity (*Shen et al., 2014*), demonstrating that newly synthesized CPEB in response to visual experience is required for neuronal plasticity.

## FUS regulates downstream RNA targets important for neuronal plasticity

FUS, a RNA binding protein associated with neurodegenerative diseases, including amyotrophic lateral sclerosis (ALS) and frontotemporal lobar degeneration (FTLD), is involved in multiple steps of RNA processing, such as transcription, splicing, transport and translation (*Lagier-Tourenne et al.,*

*2010*). Increases in FUS expression in response to mGluR activation *in vitro* suggested a role for FUS in synaptic plasticity (*Fujii et al., 2005*; *Sephton et al., 2014*). In our study, both proteomic analysis and western blots showed visual experience-dependent increases in FUS expression. Analysis of FUS knockdown indicates that FUS is required for experience-dependent structural plasticity. In addition, FUS knockdown reduces *gria1* mRNA, as reported previously (*Udagawa et al., 2015*). Together these data suggest that FUS may regulate experience-dependent structural plasticity by stabilizing *gria1* mRNA and increasing GluA1 synthesis, which may in turn enhance AMPAR-mediated glutamatergic transmission. We and others have previous shown that experience-dependent dendritic arbor structural plasticity requires AMPAR-mediated transmission (*Haas et al., 2006*; *Jablonski and Kalb, 2013*). Consistent with this, activity-dependent synthesis of GluA1 is induced in the dendrites of hippocampal neurons by dihydrexidine, a dopamine D1/D5 receptor agonist (*Smith et al., 2005*) and has been reported to be required for memory consolidation in rat (*Slipczuk et al., 2009*).

## Role of eIF3A and RPS17 in neuronal plasticity

Post-translational modifications of components of translational machinery, such as phosphorylation of eIF4F or eIF2$\alpha$, are well-studied mechanisms regulating plasticity (*Costa-Mattioli et al., 2009*; *Klann and Dever, 2004*). Our screen identifying CPPs in the visual experience-dependent nascent proteome, together with evidence that acute eIF3A and RPS17 knockdown interferes with neuronal plasticity, suggest a previously unrecognized plasticity mechanism by regulating experience-dependent *de novo* synthesis of translational machinery. eIF3A and RPS17 are part of the 43S pre-initiation complex which scans along the mRNA for the start codon to initiate protein translation after association with other initiation factors (*Hashem et al., 2013*; *Jackson et al., 2010*). Consistent with this model in which eIF3A and RPS17 affect global protein synthesis, *in vitro* studies reported broad deficits in protein synthesis with eIF3A knockdown (*Dong et al., 2004*; *Wagner et al., 2014*). An *in vivo* study in *Drosophila* showed that haploinsufficiency of RPS17 reduced protein synthesis during early embryogenesis (*Boring et al., 1989*). When we used FUNCAT to visualize the amount and distribution of NSPs with eIF3A or RPS17 knockdown in *Xenopus* tectum, we observed increased protein synthesis with eIF3A knockdown and no change in FUNCAT labeling with RPS17 knockdown after 1 hr of AHA labeling. These data are consistent with other studies suggesting non-canonical functions of eIF3A and RPS17. For instance, despite its canonical role in initiating protein translation, several studies reported additional functions for eIF3A in translational activation or repression of specific mRNAs (*Dong et al., 2004*; *Dong and Zhang, 2006*; *Lee et al., 2015*). Using photoactivatable ribonucleotide-enhanced crosslinking and immunoprecipitation (PAR-CLIP), Lee et al. reported that eIF3 complex, which contains eIF3A, binds to a specific subset of mRNA involved in cell proliferation and selectively acts as an activator or a repressor of translation for different targets (*Lee et al., 2015*). Other studies suggest that ribosomal proteins, instead of being a constitutive subunit of the 40S or 60S subunits, act as regulators for expression of a subset of gene as part of the ribosome complex or even outside of the complex (*Kondrashov et al., 2011*; *Lee et al., 2013*; *Topisirovic and Sonenberg, 2011*). Ribosomal protein L38 (Rpl38) is an example where, when it is mutated, global protein synthesis is unchanged but translation of a subset of Hox genes is affected (*Kondrashov et al., 2011*). The increase in FUNCAT labeling observed in eIF3A knockdown animals suggests that more genes may be repressed than activated by eIF3A in the *Xenopus* tectum in response to visual experience. Interestingly, 7 of our candidates were reported in the Lee et al. study to be directly bound by eIF3 complex in human 293 T cells (*Lee et al., 2015*). This suggests that synthesis of these eIF3 targets could be regulated as CPPs by eIF3A, one of the 13 subunits of eIF3 complex, with visual experience. Future studies for the specific targets of eIF3A and RPS17 in the developing brain would provide more insight into the underlying mechanism regulated by these candidates whose synthesis was increased in response to visual experience.

FUNCAT labeling shows that double knockdown of eIF3A and RPS17 increases NSPs in the neuropil but not the neuronal cell body layer. This pattern is different from knockdown of eIF3A or RPS17 alone and is not a simple combination of knocking down the individual CPPs, suggesting that eIF3A and RPS17 may function coordinately to regulate protein translation. Double knockdown of eIF3A and RPS17 exhibits stronger deficits in experience-dependent structural plasticity than single CPP knockdown. Animals with simultaneous FUS, eIF3A, and RPS17 knockdown showed the most profound deficits, including the most severe impairment in experience-dependent structural plasticity and reduced dendritic arbor growth during normal development. Moreover, triple knockdown

animals have the highest percentage of cells that failed to exhibit visual experience-dependent structural plasticity. These data suggest that inhibiting synthesis of FUS, eIF3A, and RPS17 blocks core mechanisms that neurons employ to generate visual experience-dependent structural plasticity, whereas inhibiting synthesis of one or two of these pivotal CPPs may allow others to compensate for certain functions.

In summary, we report an *in vivo* proteomic screen for experience-dependent CPPs in a behaving vertebrate, using quantitative proteomic analysis, bioinformatic predictions and *in vivo* validation with visual experience-dependent plasticity protocols including both structural and behavioral outcome measures. With the unbiased screen, we discovered novel CPPs contributing to the multifaceted plasticity events that occur in response to visual experience, and demonstrated that CPPs participating in RNA splicing and translation function in concert to mediate visual experience dependent plasticity. In total, we identified 83 CPPs that are differentially synthesized in *Xenopus* optic tectum in response to visual experience using BONCAT and DiDBiT methods. These CPPs were annotated to multiple cellular compartments, molecular functions, and biological processes, indicative of the complexity of the underlying mechanisms of visual experience-dependent plasticity. We further demonstrated that synthesis of global regulators of gene expression such as eIF3A, FUS, and RPS17 was increased in response to visual experience and required to mediate experience-dependent structural and behavioral plasticity. We propose that the dynamic synthesis of the core neuronal machinery that regulates RNA splicing and protein translation allows these proteins to serve as master regulators which control downstream effector proteins, including receptors, cytoskeletal proteins or kinases. The effector proteins then modulate synaptic structure and function, and maintain experience-induced plasticity. The master regulators could be involved in visual experience-dependent plasticity by playing their canonical roles, such as FUS, which is involved in multiple steps of RNA processing. Other CPPs, such as eIF3A and RPS17, may affect plasticity by playing their canonical roles in global protein translation, or they may play or non-canonical roles, for instance by regulating translation of a subset of targets in response to plasticity-induction protocols (*Figure 7D*).

## Materials and methods

### Key resources table

| Reagent type | Designation | Source or reference | Identifiers |
| --- | --- | --- | --- |
| Antibody | mouse-anti-CaMKIIα antibody | Novus | Cat# NB100-1983, RRID:AB_10001339 |
| Antibody | rabbit-anti-β-tubulin | Santa Cruz Biotechnology | Cat# sc-9104, RRID:AB_2241191 |
| Antibody | mouse-anti-L1CAM | Abcam | Cat# ab24345, RRID:AB_448025 |
| Antibody | rabbit-anti-BSA | Thermo Fisher Scientific | Cat# PA1-29262, RRID:AB_1956427 |
| Antibody | mouse-anti-FUS/TLS | BD Bioscience | Cat# 611385, RRID:AB_398907 |
| Antibody | rabbit-anti-eIF3A | Novus | Cat# NBP1-79628, RRID:AB_11042798 |
| Antibody | mouse-anti-RPS17 | Abnova Corporation | Cat# H00006218-M01, RRID:AB_2285214 |
| Antibody | rabbit-anti-PSMD2 | Cell Signaling Technology | Cat#14141 (This product is discontinued) |
| Antibody | mouse-anti-calmodulin | Millipore | Cat# 05–173, RRID:AB_309644 |

### Animals

Stage 46–48 albino *Xenopus laevis* tadpoles of either sex were bred in house or purchased (Xenopus Express, Brooksville, FL) and used for all experiments. Tadpoles were reared in 0.1X Steinberg's solution in a 12 hr light/dark cycle at 22–23°C until used in experiments. Animals were anesthetized in 0.02% MS-222 prior to injections or electroporation, or terminally anesthetized in 0.2% MS222. All animal protocols were approved by the Institutional Animal Use and Care Committee of The Scripps Research Institute.

### Plasmids and morpholinos

Lissamine-tagged translation-blocking antisense morpholino oligonucleotides (MO) against *Xenopus laevis* FUS, NONO, eIF3A, RPS17 were designed and generated by GeneTools with the following sequences listed. The sequence matching the start codon is underlined: Control MO (5re designed

and generated by GeneFUS MO (first 5' splice site junction) (5'-GTAATTCCTTACCGTTGGTGGC<u>CAT</u>-3'); NONO MO (5'-GTACCCTCTGTTTCCCTG<u>CAT</u>GTTT-3') (*Neant et al., 2011*); eIF3A MO (AAGTAGACCGG<u>CAT</u>TGCGGCAGATA) (*Bestman et al., 2015*); RPS17 MO (5'-TCTTTGTCCTGA-CACGTCC<u>CAT</u>GTT-3'). The sequence of FUS MO is the same as the fusMO4 previously described which can effectively block the alternative splicing of *fus* mRNA and reduce the amount of both *fus-a* and *fus-b* splice variants (*Dichmann and Harland, 2012*). FUS MO targets the first splice donor site of *fus* mRNA and is predicted to cause inclusion of intron one and reduce the amount of both *fus-a* and *fus-b* splice variants by premature termination (*Dichmann and Harland, 2012*). To evaluate MO efficacy, RT–PCR with primers covering the first four exons was conducted. MOs were dissolved in water and diluted to 0.1 mM for use in experiments. For double or triple knockdown in the structural plasticity experiments, the concentrations of individual MOs targeting candidate proteins were 0.1 mM, and control MO was 0.2 mM or 0.3 mM to match the total concentration of the MO mixture in the experimental group. To express GFP in neurons, we electroporated the optic tectum with α-actin-driven construct (pα-actin::gal4-UAS::GFP) or pSox2bd::gal4-UAS::eGFP, a construct containing the Sox2/Oct3/4 enhancer elements of the minimal FGF promoter (Sox2bd) (*Bestman et al., 2012*). For rescue experiments, we generated MO-insensitive expression construct by cloning the *Xenopus* FUS lacking the sequence targeted by FUS MO from the rescue construct, pCS108-Δ 5'UTR, a gift from Dr. Richard Harland or eIF3A and RPS17 without 5'UTR from *Xenopus* eIF3A (Open Biosystems, Clone ID# 7622710;) and *Xenopus* RPS17 (Open Biosystems, Clone ID# 5506850) respectively. The rescue constructs are designated pSox2::gal4-UAS::Δ5'UTR-FUS-t2A-eGFP, pSox2::gal4-UAS::Δ5'UTR-eIF3A-t2A-eGFP, and pSox2::gal4-UAS::Δ5'UTR-RPS17-t2A-eGFP. Plasmid concentrations we used were 0.3–1 μg/μL. To label isolated cells, we electroporated the gal4-UAS plasmids at concentration of 0.3 μg/μL supplemented with the UAS-driven eGFP (pUAS::eGFP) to increase GFP expression. Constructs and MOs were injected into the brain ventricle, then platinum electrodes were placed on each side of the midbrain and voltage pulses were applied to electroporate optic tectal cells in anesthetized stage 46 tadpoles (*Bestman et al., 2012*).

## Visual experience protocols

Two visual experience protocols that have been shown to induce plasticity in *Xenopus laevis* were employed in this study. For the proteomic analysis, we exposed animals to moving bars (1 cm width; 0.3 Hz; Luminance: 25 cd/m2) at 0.3 Hz in four cardinal directions in pseudorandom order for 10 min followed by 5 min in ambient light, repeated three times. 30–50 tadpoles were placed in a 8X3 cm tank filled with ~1 cm Steinberg's rearing solution. The bottom of the chamber was mounted with a back-projection screen. Visual stimuli were generated and presented by MATLAB 2009b (The Math-Works, Psychophysics Toolbox extensions) as previously described (*Shen et al., 2014*) and were projected on the screen using a microprojector (3M, MPro110). For western blot analysis, we used the same stimulus as above provided for either 0.5 hr or 4 hr. For the structural plasticity analysis, tadpoles were placed individually in each well of 12 well plates filled with ~1 cm Steinberg's rearing solution in a box wrapped with foil for 4 hr and then exposed animals to 4 hr visual experience in which rows of LEDs were turned on and off sequentially at a frequency of 0.2 Hz as previously described. This visual stimulus protocol consistently induces visual experience-dependent dendritic arbor structural plasticity, detected with *in vivo* time-lapse imaging of individual GFP-expressing neurons (*Bestman and Cline, 2008*; *Haas et al., 2006*; *Li et al., 2011*; *Sin et al., 2002*). Behavioral plasticity was tested after 4 hr of visual experience composed of moving bars.

## BONCAT for DiDBIT

500 mM AHA (L-azidohomoalanine, 500 mM, pH7.4, Clickchemistry tools) colored with ~0.01% fast green was injected into the tectal ventricle of anesthetized stage 47/48 tadpoles. Animals recovered from anesthesia for 0.5 hr and then were exposed to VE for 0.5 hr followed by 4 hr ambient light before dissecting out their midbrains. We dissected midbrains from 1200 to 1500 stage 47/48 tadpoles for each experimental group in order to yield about 15 mg protein in two separate experiments. Brains were homogenized in phosphate-buffered saline (PBS) containing 0.5% SDS and protease inhibitors (PI; Roche, complete ULTRA Tablets, Mini, EDTA-free Protease Inhibitor cocktail tablets) followed by sonication with a probe sonicator. Samples were boiled for 5 min and small aliquots were taken to measure protein concentration using the BCA Protein Assay Kit (Thermo Fisher

Scientific, 23227). 5–10 μg of the sample was saved as total protein sample, while the rest was used for the following click reaction. For each 400 μL reaction, 1.5 mg of total protein was used with 1.7 mM Triazole ligand (Invitrogen) in 4:1 tBuOH/DMSO (Sigma), 50 mM CuSO4 (Sigma), 5 mM Biotin Alkyne (Invitrogen) and 50 mM TCEP (Sigma) added in sequence. The reaction proceeded for 1–2 hr at room temperature. Excess reagents were removed with methanol/chloroform/water precipitation.

## DiDBIT

Precipitated proteins from 10 click reactions were combined, air-dried and resuspended in 100 μL of 0.2% ProteaseMAX (Promega, Madison, WI) and then 100 μL of 8M urea was added. The solution was reduced with 5 mM TCEP for 20 min at 37°C, and then reduced with 10 mM IAA for 20 min in the dark at room temperature. Next, 150 μL of 50 mM ammonium bicarbonate and 2.5 μL of 1% ProteaseMAX were added prior to the addition of 200 μg trypsin. The sample was digested for three hours at 37°C in a shaking incubator. The peptides were desalted as previously described (*Villén and Gygi, 2008*) and dried with a speed-vac prior to AHA-peptide enrichment. The peptides were resuspended in 1 ml PBS and incubated with 200 μL washed Neutravidin beads (Pierce) at room temperature for 2 hr. Beads were washed with PBS and the peptides were eluted with elution buffer (0.1% TFA/0.1% formic acid/70% acetonitrile in $H_2O$). After drying the eluted AHA peptides with a speed-vac, the peptides were labeled with dimethyl tags as previously described (*Boersema et al., 2009*).The control sample was labeled with the light tag and VE sample was labeled with the heavy tag. Unmodified peptides from the flow-through after AHA enrichment were labeled in an identical manner.

## Multidimensional protein identification technology (MudPIT)

Labeled peptides from the control and VE samples were mixed 1:1 based on the protein quantification of the starting material. Next, they were pressure-loaded onto a 250 μm i.d capillary with a kasil frit containing 2 cm of 10 μm Jupiter C18-A material (Phenomenex, Ventura, CA) followed by 2 cm 5 μm Partisphere strong cation exchanger (Whatman, Clifton, NJ). This loading column was washed with buffer A. After washing, a 100 μm i.d capillary with a 5 μm pulled tip packed with 15 cm 4 μm Jupiter C18 material (Phenomenex, Ventura, CA) was attached to the loading column with a union and the entire split-column (loading column–union–analytical column) was placed in line with an Agilent 1100 quaternary HPLC (Palo Alto, CA). The sample was analyzed using MudPIT, which is a modified 12-step separation described previously (*Washburn et al., 2001*). The buffer solutions used were buffer A, 80% acetonitrile/0.1% formic acid (buffer B), and 500 mM ammonium acetate/5% acetonitrile/0.1% formic acid (buffer C). Step 1 consisted of a 90 min gradient from 0–100% buffer B. Steps 2–11 had the following profile: 3 min of 100% buffer A, 5 min of X% buffer C, a 10 min gradient from 0–10% buffer B, and a 105 min gradient from 15–45% buffer B. The buffer C percentages (X) were 20, 30, 40, 50, 60, 70, 60, 100%, respectively for the 11-step analysis. In the final two steps, the gradient contained: 5 min of 100% buffer A, 5 min of 90% buffer C plus 10% B, a 10 min gradient from 0–15% buffer B, and a 105 min gradient from 15–100% buffer B. As peptides eluted from the microcapillary column, they were electrosprayed directly into an Elite mass spectrometer (ThermoFischer, Palo Alto, CA) with the application of a distal 2.4 kV spray voltage using the rapid scan settings previously published (*Michalski et al., 2012*). Applications of mass spectrometer scan functions and HPLC solvent gradients were controlled by the Xcalibur data system. MudPIT analysis was performed twice for two AHA peptide samples from two biological samples. The flow-through samples were analyzed by three MudPITs for each biological sample (i.e. three technical replicates). The MS data from the technical replicates were combined into one dataset prior to data analysis. The raw MS spectra files and DTASelect files have been uploaded to www.proteomexchange.org (PXD008694) via MassIVE (MSV000081728).

## Analysis of tandem mass spectra

Both MS1 and MS2 (tandem mass spectra) were extracted from the XCalibur data system format (.RAW) into MS1 and MS2 formats using in house software (RAW_Xtractor) (*McDonald et al., 2004*). Both binary and source codes are available at https://github.com/robinparky/rawconverter/ (*Park, 2018*; copy archived at https://github.com/elifesciences-publications/rawconverter). MS2 spectra remaining after filtering were searched with the Prolucid Software (*Xu et al., 2015*)

separately against three different databases: UniProt_Xenopus_laevis_01-23-2015, Xenbase_Xenopus_laevis_05-29-2014 ([http://www.xenbase.org/](http://www.xenbase.org/), RRID:SCR_003280), and PHROG_07-01-2014 (*Wühr et al., 2014*). Each database was concatenated to a decoy database in which the sequence for each entry in the original database was reversed (*Peng et al., 2003*). All searches were parallelized and performed on a Beowulf computer cluster consisting of 100 1.2 GHz Athlon CPUs (*Sadygov et al., 2002*). No enzyme specificity was considered for any search. The following modifications were searched for a static modification of 57.02146 on cysteine and a differential modification of 523.2749 on methionine for AHA. The 'light' and 'heavy' dimethylation of $NH_2$-terminus and lysine were searched (*Boersema et al., 2009*). Prolucid results were assembled and filtered using the DTASelect (version 2.0) program (*Cociorva et al., 2007*; *Tabb et al., 2002*). DTASelect 2.0 uses a linear discriminant analysis to dynamically set XCorr and DeltaCN thresholds for the entire dataset to achieve a user-specified false discovery rate (FDR). In addition, the modified peptides were required to be fully tryptic, less than 5ppm deviation from peptide match, and a FDR at the spectra level of 0.01. The FDRs are estimated by the program from the number and quality of spectral matches to the decoy database. For all datasets, the protein FDR was <1% and the peptide FDR was <0.5%. Census was employed to generate heavy/light peptide ratios using the MS1 files and confident identifications from DTASelect (*Park et al., 2008*). The average AHA peptide ratio for each protein was shifted to 1:1 based on the median ratio of the quantified unmodified peptides from the flow-through for each experiment. The normalized fold changes VE:control (V/C) in AHA labeling from three databases were averaged to represent the fold change in each experiment (*Supplementary file 2*). The final fold change in AHA labeling is the average of normalized fold changes of two independent experiments (*Supplementary file 2*, *Table 1*).

## BONCAT for western blot

For analysis of AHA-biotin tagged proteins by western blots, animals received AHA injections and visual experience protocols as described for proteomic analysis. Protein homogenates from 30 to 50 tecta from each experimental group as described above and comparable protein amounts from each group were added to the click reaction together with biotinlyated BSA (BioVision, 7097–5), which served as the internal control. The click reaction and protein precipitation were performed as described for the proteomic analysis. The dried protein pellets were suspended in 100 µL of 6M Urea/25 mM ammonium bicarbonate/0.5% SDS in PBS with vortexing for 10–20 min or until the pellet was dissolved. We added 50–100 µL washed Neutravidin beads (Pierce, 29200) and added PBS to a final volume of 1000 µL. Samples were subjected to head-over-head rotation for at least 2 hr at room temperature. Beads were rinsed with PBS before incubation in 1% SDS for 15 min followed by two washes with PBS. Finally, after one last wash with water, 2X sample buffer was added to the remaining Neutravidin beads, boiled for 10–15 min. After the solution cooled down, we ran the protein sample on a SDS-polyacrylamide gel within 24 hr.

## Bioinformatic analysis

Gene Ontology (GO) analysis was performed using the gene symbols to search against the human database in PANTHER (version 11.1) (*Mi et al., 2016*). *Figure 1B–D* represents the percent of gene hits with annotated PANTHER protein classes against total number protein class hits. Note that some genes were assigned to multiple PANTHER protein classes. To retrieve statistically enriched GO terms and to construct protein interaction networks, we used both human and mouse STRING database (version 10.0) (*Szklarczyk et al., 2015*). For STRING network, we used the high confidence (0.7) as our minimum required interaction score and included active interaction sources from experiments, databases, co-expression, neighborhood, gene fusion and co-occurrence. We used the SynProt classic and PreProt databases from SynProt Portal ([www.synprot.de](www.synprot.de)) to examine if our candidate proteins were annotated to synaptic junctions which is detergent-resistant and presynaptic localizations including synaptic vesicle, cytomatrix, and active zone (*Pielot et al., 2012*).

## Western blot and immunocytochemistry

To evaluate knockdown or overexpression (OE) of FUS, eIF3A or RPS17 by western blot, stage 47/48 tadpole midbrains were electroporated with 0.1 mM MOs or 1–2 µg/µL plasmids and dissected two days later. Experimental and paired control samples were prepared and processed side by side.

Tissues were homogenized in RIPA buffer and boiled for 5 min before brief sonication. After measuring protein concentration with BCA Protein Assay Kit, 2X sample buffer was added to the sample and boiled for 5–10 min. 5–10 µg of lysate was loaded onto an Mini-Protean TGX precast gels (Bio-Rad) and proteins were transferred to a nitrocellulose membrane with Trans-Blot Turbo transfer system (BioRad). The membrane was incubated in 5% non-fat milk/0.05% Tween-20 (Sigma) in TBS for an hour for blocking, and then transferred to primary antibodies diluted in blocking solution and incubated 1–2 overnight at 4°C. After three brief washes with 0.05% Tween-20 in TBS, membranes were transferred to secondary antibodies, goat anti-mouse or goat anti-rabbit HRP-conjugated secondary (BioRad), diluted in blocking solution for an hour at room temperature. Blots were rinsed and incubated with HRP-linked mouse/rabbit/goat IgG (BioRad). The Pierce ECL western Blot substrate (Thermo Fisher Scientific, 32209) was used to visualize labeling. For quantification of western blots, different exposure periods were used for the same blots to avoid saturation. The blots were scanned and band intensities were measured from non-saturating exposures with ImageJ. For the BONCAT samples which had biotinylated BSA spiked in, the band intensity of each candidate protein was first normalized to its BSA loading control band (which was obtained after stripping the same membrane) and then that value was normalized to the β-tubulin loading control band from input/total protein samples. For other samples, the band intensity of each candidate protein was normalized to its β-tubulin loading control band. For comparison between VE and control, we calculated the ratios of normalized intensity values (VE/control or control/control) in each set of paired conditions. We use total β-tubulin as a loading control, based on our previous study showing that total β-tubulin is stable in response to visual experience, whereas newly synthesized β-tubulin increases in response to visual experience (*Shen et al., 2014*). Outliers, defined as those with ratio of normalized intensity (VE/control) greater that two SD from the mean, were excluded from the analysis of total proteins.

The following antibodies were used in this study. Mouse-anti-CaMKIIα antibody (Novus, NB100-1983), rabbit-anti-β-tubulin (Santa Cruz, sc-9104), mouse-anti-L1CAM (Abcam, ab24345), rabbit-anti-BSA (Thermo Fisher Scientific, PA1-29262), mouse-anti-FUS/TLS (BD Bioscience, 611385), rabbit-anti-eIF3A (Novus, NBP1-79628), mouse-anti-RPS17 (Novus, H00006218-M01), rabbit-anti-PSMD2 (Cell Signaling Technology, 14141), mouse-anti-calmodulin (Millipore, 05–173), goat-anti-mouse IgG (H + L)-HRP conjugate (BioRad, 172–1011), and goat-anti-rabbit IgG(H + L)-HRP conjugate secondary antibodies (BioRad, 172–1019).

To evaluate RPS17 expression using immunocytochemistry, HEK 293 T cells were grown on 15 mm coverslips (Corning, 354087) in 24 well plates with DMEM media (Gibco) supplemented with 20% fetal bovine serum and Penicillin/Streptomycin solution (Gibco). When the cells grew to 70–80% confluence, the culture medium was changed to DMEM and the plasmids mixed with Lipofectamine 2000 (Invitrogen) were added to each well. Cells were transfected with either pSox2::gal4-UAS::t2A-eGFP (control group) or pSox2::gal4-UAS:: Δ5'UTR-RPS17-t2A-eGFP (RPS17 expression group). After two days of incubation at 37°C in a $CO_2$ incubator, cells were fixed with 4% paraformaldehyde (PFA, pH 7.4) for 15 min at room temperature, washed with ice-cold PBS twice, permeabilized with 0.3% Triton-X 100 in PBS (PBST) for 15 min, and then blocked with 1% BSA in PBST for 1 hr at room temperature. Coverslips were then transferred to Mouse-anti-RPS17 primary antibody (same as above) overnight at 4°C, followed by 2 hr in donkey-anti-mouse Alexa Fluor 647 (Life Technologies, A-31571) at room temperature. Coverslips were counterstained with DAPI for 15 min before mounting. Samples were cleared and mounted in 50% glycerol/6M Urea and imaged on a Nikon C2 confocal microscope with a 20X (0.75 NA) lens. To test for RPS17 expression in HEK 293 T cells, we quantified the average RPS17 labeling intensity per unit area within the masks of regions of interest (ROIs), created based on GFP expression. RPS17 labeling intensity in GFP⁻ ROI lacking GFP expression was used for normalization before combining results from different experiments.

## Real Time-PCR

To validate the effect of FUS MO knockdown, total RNA was isolated from tadpole midbrains dissected two days after control MO or FUS MO were electroporated at stage 46 using Trizol (Life Technologies). cDNA was synthesized using SuperScript III (Life Technologies) with oligo-dT primer. Genes of interest were amplified by PCR using GoTag green master mix (Promega). Primer sets used in this study including *rps13* (*Thompson and Cline, 2016*), *fus-a* (*Dichmann and Harland, 2012*), *gria1*, and *gria2* are listed in *Supplementary file 7*. Different amplification cycles, ranging from 25 to 30 cycles, were used for different genes to avoid amplifications reaching plateau. For

quantification of DNA fragments amplified by PCR, different exposure periods were used for the same gels to avoid saturation. The band intensities were measured from non-saturating exposures with ImageJ and normalized to the *rps13* loading control band of each group. Inter-group differences were assessed by one-tailed Student's t test.

## In vivo time-lapse imaging of structural plasticity

The optic tecta of stage 46 animals were electroporated with plasmids and MOs for knockdown or rescue experiments as described below. Knockdown experiments: 0.3 µg/µL pα-actin::gal4-UAS:: GFP and 0.1 mM control/FUS/eIF3A/RPS17 MOs; 0.3 µg/µL pα-actin::gal4-UAS:: GFP and 0.2 mM control/0.1 mM eIF3A + 0.1 mM RPS17 MOs; 0.3 µg/µL pα-actin::gal4-UAS:: GFP and 0.3 mM control/0.1 mM FUS +0.1 mM eIF3A + 0.1 mM RPS17 MOs. Rescue experiments: Control MO: pSox2bd::gal4-UAS::eGFP alone or supplemented with 0.25 ug/µL pUAS::eGFP and 0.1 mM control MO; FUS/eIF3A/RPS17 MO: 0.3 µg/µL pSox2bd::gal4-UAS::eGFP with 0.1 mM FUS/eIF3A/RPS17 MO; FUS MO +FUS: 0.5 ug/µL pSox2::gal4-UAS::Δ5'UTR-FUS-t2a-eGFP supplemented with 0.25 ug/µL pUAS::eGFP and 0.1 mM FUS MO; eIF3A MO +eIF3A: 0.5 µg/µL pSox2::gal4-UAS::Δ5'UTR-eIF3A-t2a-eGFP supplemented with 0.25 µg/µL UAS::eGFP and 0.1 mM eIF3A MO; RPS17 MO +RPS17: 0.5 µg/µL pSox2::gal4-UAS::Δ5'UTR-RPS17-t2a-eGFP supplemented with 0.25 µg/µL pUAS::eGFP and 0.1 mM RPS17 MO. Animals were screened for those with sparsely transfected and well-isolated cells under an epifluorescent microscope one day after electroporation. Two days after electroporation, single neurons in intact animals were imaged on a custom-built two photon microscope with a 20X (0.95 NA) water immersion lens at 2–3.5X scan zoom. The dendrites of single neurons were traced and reconstructed using the Vaa3D-Neuron 2.0: 3D neuron paint and tracing function in Vaa3D (http://vaa3d.org) with manual correction and validation of the tracing (*Peng et al., 2010*). Total dendritic branch length (TDBL) was quantified and growth rates were determined as changes in TDBL after 4 hr in the dark or 4 hr with visual experience. Two-tailed paired Student's t test were used to compare between two matched pairs of TDBL after 4 hr in the dark and TDBL after 4 hr visual experience of the same neuron. All samples were imaged in parallel using the same image acquisition parameters. Power analyses of the control datasets indicate that the structural plasticity studies are properly powered (power ranges from 0.75 to 0.99).

## FUNCAT and quantification

The optic tecta of stage 46 animals were electroporated with 0.1 mM control MO, 0.1 mM eIF3A MO, 0.1 mM RPS17 MO, or 0.1 mM eIF3A + 0.1 mM RPS17 MO. Two days after electroporation, 500 mM AHA colored with ~0.01% fast green was injected into the midbrain ventricle of anesthetized tadpoles. One hour after AHA was injected into the ventricle, midbrains were dissected and fixed with 4% PFA (pH 7.4). Samples were processed for click chemistry, washed several times and mounted in clearing solution of 50% glycerol/6 M Urea before being imaged with an Olympus FluoView500 confocal microscope with a 20X (0.8 NA) oil immersion lens. Fluorescence intensity of AHA labeling in the neuronal cell body layer or in the neuropil was quantified in single optical sections from confocal z-series through the brain using custom applications created in MATLAB 2009b (The MathWorks, Psychophysics Toolbox extensions). Measurements from the neuronal cell body layer were made between 20–30 pixels (12.4 to 18.6 µm) to the left and right of the midline and measurements from the neuropil were made between 80–100 pixels (49.6 to 68.2 µm) to the left and right of the midline. Further details about FUNCAT labeling and quantification of AHA labeling can be found in (*Liu and Cline, 2016*).

## Visual avoidance assay

The visual avoidance assay was conducted as reported (*Dong et al., 2009*; *Shen et al., 2014*). Briefly, 4–5 tadpoles were placed in a 8X3 cm tank filled with ~1 cm Steinberg's rearing solution. The bottom of the chamber was mounted with a back-projection screen. Visual stimuli were projected on the screen using a microprojector (3M, MPro110). Videos of tadpoles illuminated by IR LEDs were recorded with a Hamamatsu ORCA-ER digital camera. Visual stimuli were generated and presented by MATLAB 2009b (The MathWorks, Psychophysics Toolbox extensions). Randomly positioned moving spots of 0.4 cm diameter were presented in pseudorandom order for 60 s. Visual avoidance behavior was scored as a change in swim trajectory or speed and plotted as an avoidance

index (AI), the ratio of avoidance responses to first 10 encounters with an approaching visual stimulus. Animals in which more than 50% of turning events were independent of an encounter with visual stimuli were not included for further analysis.

## Statistical tests

All data are presented as mean ±SEM based on at least three independent experiments except *Figure 2*, *Figure 4C–D*, *Figure 5B–D*, *Figure 6B–D*, and *Figure 4—figure supplement 1*. Data are considered significantly different when p values are less than 0.05. The nonparametric Mann-Whitney and one-tailed or two-tailed Student's t test were used for comparisons of two groups. The two-tailed paired Student's t test was used to compare between two matched pairs in *Figure 4C–D*, *Figure 5B–D*, *Figure 6B–D*, and *Figure 4—figure supplement 1*. The Steel-Dwass test was used for nonparametric multiple comparisons with control as stated. See *Supplementary file 9* for further information about choices of statistical tests and the p values for each figure. JMP 11 statistics software (SAS institute Inc.) was used for all statistics analysis. All samples were prepared and analyzed in parallel, blind to treatment.

## Acknowledgements

We thank Dr. Richard Harland for sharing the FUS MO and FUS rescue construct, and Dr. Marc Moreau for sharing the NONO MO. We thank members of Cline lab, Drs Ben Cravatt and Anton Maximov for discussion and critical comments on the manuscript. This work was supported by grants from the US National Institutes of Health (EY011261), Salk NEI Core (P30 EY019005), a fellowship from the Helen Dorris Foundation to H-HL, support from DartNeuroScience,LLC and an endowment from the Hahn Family Foundation to HTC, and NIH grants 5R01MH067880 and 5 R01 MH100175 to JRY.

## Additional information

### Funding

| Funder | Grant reference number | Author |
|---|---|---|
| National Institutes of Health | EY011261 | Han-Hsuan Liu<br>Lucio Schiapparelli<br>Wanhua Shen<br>Hollis T Cline |
| National Institutes of Health | EY019005 | Han-Hsuan Liu<br>Lucio Schiapparelli<br>Hollis T Cline |
| DartNeuroScience LLC | | Han-Hsuan Liu<br>Hollis T Cline |
| National Institutes of Health | MH099799 | Han-Hsuan Liu<br>Lucio Schiapparelli<br>Hollis T Cline |
| National Institutes of Health | MH067880 | Daniel B Mcclatchy<br>John R Yates III |
| National Institutes of Health | MH100175 | Daniel B Mcclatchy<br>John R Yates III |

The funders had no role in study design, data collection and interpretation, or the decision to submit the work for publication.

### Author contributions

Han-Hsuan Liu, Conceptualization, Formal analysis, Investigation, Methodology, Writing—original draft, Writing—review and editing; Daniel B McClatchy, Formal analysis, Investigation, Methodology, Writing—original draft; Lucio Schiapparelli, Investigation, Methodology, Writing—original draft; Wanhua Shen, Conceptualization, Investigation, Methodology; John R Yates III, Software,

Supervision, Methodology; Hollis T Cline, Conceptualization, Supervision, Funding acquisition, Writing—original draft, Writing—review and editing

### Author ORCIDs
Han-Hsuan Liu http://orcid.org/0000-0002-5330-1689
Daniel B McClatchy http://orcid.org/0000-0002-0288-5645
Hollis T Cline http://orcid.org/0000-0002-4887-9603

### Ethics
Animal experimentation: All animal protocols (#08-0083-4) were approved by the Institutional Animal Use and Care Committee of The Scripps Research Institute.

### Decision letter and Author response
Decision letter https://doi.org/10.7554/eLife.33420.039
Author response https://doi.org/10.7554/eLife.33420.040

## Additional files

### Supplementary files
• Supplementary file 1. Table of proteins identified in global brain proteome and nascent proteome. Related to *Figure 1*. List of genes identified by searching MS/MS spectra against three different databases, including Uniprot *Xenopus laevis* database, Xenbase, and PHROG and then converted to human homologs by gene symbol. From two independent experiments, we identified 4833 unmodified proteins in the global brain proteome (sheet 1) and 835 AHA-labeled NSPs in the nascent proteome (sheet 2).
DOI: https://doi.org/10.7554/eLife.33420.022

• Supplementary file 2. Normalized fold change in AHA labeling searched against three different databases. Related to *Table 1*. The normalized fold changes visual experience (VE):control (V/C) in AHA labeling from two independent experiments by searching MS/MS spectra against three different databases were listed. The final fold changes in AHA labeling is from the averaged of two independent experiments are also shown in *Table 1*. CPPs are shaded in red if they increased in both experiments, in pink if increased in one of the experiments, in green if decreased in both experiments, and in light green if decrease in on of the experiments.
DOI: https://doi.org/10.7554/eLife.33420.023

• Supplementary file 3. Table of proteins identified in VE-dependent nascent proteome. Related to *Table 1*. 83 CPPs are annotated to a variety of cellular compartments, molecular functions, and biological processes by using the PANTHER database.
DOI: https://doi.org/10.7554/eLife.33420.024

• Supplementary file 4. The enriched GO annotation in biological process of 83 CPPs. Related to *Table 1*. We identified the enriched GO annotation using both human and mouse protein interaction databases provided by STRING.
DOI: https://doi.org/10.7554/eLife.33420.025

• Supplementary file 5. List of the PANTHER protein classes in the global brain proteome, nascent proteome, and VE-dependent nascent proteome. Related to *Figure 1*. Proteins were annotated using PANTHER. The breakdown of the 'others' category in the pie charts in *Figure 1B–D* is included.
DOI: https://doi.org/10.7554/eLife.33420.026

• Supplementary file 6. Summary of synaptic localizations of CPPs. Related to *Table 1*. The synaptic localizations of 83 CPPs are derived from SynProt and PreProt databases.
DOI: https://doi.org/10.7554/eLife.33420.027

• Supplementary file 7. RT-PCR primer oligonucleotides used for this study. Related to Experimental Procedures. Forward and reverse primer sequences that were used for quantification of gene expression are shown.
DOI: https://doi.org/10.7554/eLife.33420.028

• Supplementary file 8. Overlap of 5 hr optic tectal nascent proteome with the 24 hr whole brain nascent proteome. List of 992 proteins from the *Shen et al. (2014)* study of AHA-labeled NSPs from tadpole brains labeled over a 24 hr period, generated by searching against the Uniprot database. We compared the 5 hr optic tectal nascent proteome (*Supplementary file 1*), searched against the Uniprot database, with the 24 hr whole brain nascent proteome, and highlighted the overlapping proteins. The dataset for the 24 hr whole brain nascent proteome is available at http://proteomecentral.proteomexchange.org/cgi/GetDataset?ID=PXD008659.

DOI: https://doi.org/10.7554/eLife.33420.029

• Supplementary file 9. Statistical table. For each statistical test run in the study, the data structure, type of statistical test, sample size, p-value and power are listed.

DOI: https://doi.org/10.7554/eLife.33420.030

• Transparent reporting form

DOI: https://doi.org/10.7554/eLife.33420.031

### Major datasets

The following dataset was generated:

| Author(s) | Year | Dataset title | Dataset URL | Database, license, and accessibility information |
|---|---|---|---|---|
| Han-Hsuan Liu, Daniel B Mcclatchy, Lucio Schiapparelli, Wanhua Shen, John R Yates, Hollis T Cline | 2018 | Visual Conditioning-Dependent Nascent Proteome in *Xenopus* optic tectum | http://proteomecentral. proteomexchange.org/ cgi/GetDataset?ID= PXD008694 | Publicly available at ProteomeXchange (accession no. PXD00 8694) |

The following previously published datasets were used:

| Author(s) | Year | Dataset title | Dataset URL | Database, license, and accessibility information |
|---|---|---|---|---|
| Wanhua Shen, Han-Hsuan Liu, Lucio Schiapparelli, Daniel B Mcclatchy, John R Yates, Hollis T Cline | 2014 | Xenopus brain proteome | http://proteomecentral. proteomexchange.org/ cgi/GetDataset?ID= PXD008659 | Publicly available at ProteomeXchange (accession no. PXD00 8659) |
| Wühr M, Freeman RM, Presler M, Horb ME, Peshkin L, Gygi S, Kirschner MW | 2014 | Deep Proteomics of the Xenopus laevis Egg using an mRNA-derived Reference Database | https://www.ebi.ac.uk/ pride/archive/projects/ PXD000926 | Publicly available at ProteomeXchange (accession no. PXD000926) |

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
