## [Decision Letter]

Thank you for submitting your article "Role of the Visual Experience-Dependent Nascent Proteome in Neuronal Plasticity" for consideration by *eLife*. Your article has been reviewed by three peer reviewers, one of whom is a member of our Board of Reviewing Editors and the evaluation has been overseen by a Senior Editor. The following individual involved in review of your submission has agreed to reveal her identity: Elly Nedivi (Reviewer #3).

The reviewers have discussed the reviews with one another and the Reviewing Editor has drafted this decision to help you prepare a revised submission.

Summary:

The authors of this manuscript have developed the Xenopus optic tectum as a model system for studying cellular mechanisms of activity-dependent neuronal plasticity. Previously they have established that de novo protein synthesis is required for experience-dependent structural and functional plasticity of optic tectum neurons in vivo and they have reported their development of a high sensitivity mass spec-based assay for nascent protein synthesis in this system. Here they now use this method to identify proteins newly synthesized in the optic tectum in response to visually evoked activity. All of the reviewers agreed that overall the biochemical experiments were performed to a high standard, and that the data resulting from this technically demanding effort, which identified 83 proteins showing activity-dependent increases in protein synthesis, would be of significant interest as a resource to a broad group of readers interested in mechanisms of neuronal plasticity.

The second half of the manuscript tests the function of three activity-regulated candidates from this list via morpholino-based knockdown in optic tectum *in vivo*. Although these experiments are useful as validation of the biochemical screen, there was substantial debate among the reviewers about the significance of the findings. The authors claim these data are important because "they indicate that de novo synthesis of proteins required for RNA splicing and protein synthesis are themselves required for plasticity, introducing a new locus for control of experience-dependent plasticity." The simple fact that regulators of splicing and translation are among the gene products that can be regulated at the level of translation is interesting but conceptually incremental from the work the authors have already done to establish the importance of translation for neuronal plasticity in this system, and notably the authors have previously shown that other kinds of RNA regulatory proteins (e.g. the RNA binding protein CPEB) show de novo protein synthesis-dependent effects on optic tectum plasticity.

In response to a previous review that raised similar concerns about significance of the candidate testing, the authors wrote that they "demonstrate that it is their activity-dependent synthesis that is required for experience-dependent plasticity, not simply the presence of the candidate protein." Despite the fact we largely agreed (with caveats) that the authors show that knockdown of eIF3A, RPS17, and FUS selectively impair optic tectum neuron structure or function selectively during the experience-dependent phase, given the way this study was performed those effects are due to knockdown of the candidate proteins over the 2 days preceding experience and therefore cannot be selectively attributed to loss of the ~20% change in expression of these proteins following visual experience. Although the reviewers debated possible additional experiments that could strengthen the second section, many options were seen as beyond the scope of the current study, and one reviewer felt the new suggestions did not change the fundamental argument about the timing of the function of the RNA regulators tested here.

Essential revisions:

Given this analysis, we propose that the authors revise the submission as a Tools and Resource paper rather than a Research Article. By centering on the dataset rather than the functional analysis the manuscript will reflect the most impactful points. We also offer a series of textual revisions below that will strengthen the paper.

We appreciate that this is a large list but feel that all of the following points can and should be addressed with text revisions.

1) The authors determined which of their 83 candidates overlapped with FMRP targets and with the SAFARI database of autism spectrum-associated genes, but they did not report which of these genes are novel candidate plasticity genes and which had been identified in previous screens for candidate plasticity genes. What percentage of the proteins that they identified were novel, and what percentage were already known to be activity-regulated? This is important for establishing the significance of the unbiased screen.

2) The mRNAs identified by FRMP Clip can't really be called ASD genes. This should be reserved for proteins that actually are implicated in autism (the SFARI database), not indirectly proposed as candidates because they are regulated by FMRP. Indeed, since a reasonable number of FMRP clipped targets are in the SFARI database, the result of an enrichment in CLIP targets, but not SFARI targets suggests enrichment for a set of FMRP targets that are not implicated in autism.

3) In Figure 4 is the image shown from a control cell?

4) There are several issues with Figure 2. Firstly, representative images of western blots should be included. Figure 2 are very unclear. Why is eIF3A only shown in Figure 2 and not 2B? The statistical analysis should use ANOVA and not t-test since multiple samples are being compared. Also, there are no lines to indicate which samples are being compared to each other so the stars above each column for p values are ambiguous.

5) For the proteomic screen, it should be explicitly stated how many animals were used and how many repeats were done in the main text or in the Figure legends. This information is suggested in Table 1 and can be found by reading the Materials and methods section very carefully but it is too important to be difficult to figure out.

6) Figure 6 has no error bars or statistical analysis. Were multiple cells imaged per animal? Again, what is the n? ANOVA should be used instead of a t-test. The way the data is presented in the Figure legend of 7B,C is appropriate and all the legends in the manuscript should include this type of information.

7) In Subsection “Blocking both protein translation and RNA splicing has profound effects on visual experience-dependent structural and behavioral plasticity” please clarify how dendritic growth rate is measured.

8) There are confusing statements about Fus alternative splicing without any explanation. Why was the splice junction targeted, what are the splice variants, what splice is remaining here? This should be explained.

9) While the observation of changes in growth were only down in the electroporated cells, changes in behavior presumably require electroporation of a large percentage of the tectal neurons. This should be discussed or explained.

10). Based on the time points that the authors have used, only a subset of proteins that are likely to be important for translation-dependent plasticity have probably been identified and this would miss many proteins that are synthesized quickly and then degraded, such as Arc and BDNF. Indeed, this same group examined a subset of selected proteins for increases in AHA-labeling acutely after learning (30 minutes of training, 2 hours afterwards), somewhat shorter than the time course examined here (4 hours afterwards). Relatively strong increases in CAMKII, GAD65, MEK1 and HDAC were observed. These were not replicated in the present study. Indeed, they still use CAMKII as their positive control in Figure 2 despite the lack of a change in their proteomics. Discussion of this is warranted.

11) There is no evidence that there is a specific increase in the translation rate of the proteins identified, as opposed to an increase in the amount of protein made due to increased transcription followed by translation. An increase in transcription and basal translation would still lead to an increase in AHA-labeling, particular in a 4 hour labelling period. Again, this needs to be discussed.

12) Knockdowns of eIF3A and Rps17 are likely to decrease overall protein synthesis and this is a confound to the interpretation of these experiments. Indeed, the authors acknowledge this (haploinsufficiency of Rps17 causes small *Drosophila*; large effects on translation with knockdowns of eIF3A). However, whether or not the morpholinos cause a change in overall translation is not measured by the applicants (i.e. AHA labeling with immunocytochemistry comparing knockdown cells to adjoining cells without the knockdown). Since the plasticity requires protein synthesis, it is not clear that these experiments are not just the equivalent of adding anisomycin. A more general problem is that knocking down overall synthesis for a long period of time is not equivalent to knocking down the small increase in newly synthesized protein, since much larger changes in the proteome are induced by the manipulation than occur during memory. Both of these caveats should be discussed.

13) The methods in immunoblotting state that "Outliers, defined as those with AI greater that two SDs from the mean, were excluded from the analysis of total proteins". I assumed that many animals were pooled for these experiments (30-50 tecta), so I do not understand this statement.

14) Figure 2—figure supplement 1 is never mentioned in the text.

15) It is not clear what statistical test was used to sow that the effect of the double and triple Kos were larger than the single Kos as 7 there is no statistical comparison between the CMO and triple knockout. What was the definition of a 'non-plastic' cell. In Figure 7, the p value for the 'non-significant' result needs to be given as it appears to be close to significance. This should be given for all cells, or at least a general statement such as non-significant, p>0.5. P values less than or close to 0.1 are of borderline significance and in these cases, the actual p values should be given.

---

## [Author Response]

Essential revisions:Given this analysis, we propose that the authors revise the submission as a Tools and Resource paper rather than a Research Article. By centering on the dataset rather than the functional analysis the manuscript will reflect the most impactful points. We also offer a series of textual revisions below that will strengthen the paper.We appreciate that this is a large list but feel that all of the following points can and should be addressed with text revisions.1) The authors determined which of their 83 candidates overlapped with FMRP targets and with the SAFARI database of autism spectrum-associated genes, but they did not report which of these genes are novel candidate plasticity genes and which had been identified in previous screens for candidate plasticity genes. What percentage of the proteins that they identified were novel, and what percentage were already known to be activity-regulated? This is important for establishing the significance of the unbiased screen.

Our CPP dataset did not overlap with previously reported immediate early genes or candidate plasticity genes induced by strong induction protocols, however some CPPs from our screen were recently identified in two screens for stimulus-responsive genes and proteins in mouse. We have added this to the Discussion section.

2) The mRNAs identified by FRMP Clip can't really be called ASD genes. This should be reserved for proteins that actually are implicated in autism (the SFARI database), not indirectly proposed as candidates because they are regulated by FMRP. Indeed, since a reasonable number of FMRP clipped targets are in the SFARI database, the result of an enrichment in CLIP targets, but not SFARI targets suggests enrichment for a set of FMRP targets that are not implicated in autism.

We changed Table 1, its legend and the text in the manuscript to distinguish ASD genes and FMRP targets.

3) In Figure 4 is the image shown from a control cell?

Yes. This is now mentioned in the text.

4) There are several issues with Figure 2. Firstly, representative images of western blots should be included. Figure 2 are very unclear. Why is eIF3A only shown in Figure 2 and not 2B? The statistical analysis should use ANOVA and not t-test since multiple samples are being compared. Also, there are no lines to indicate which samples are being compared to each other so the stars above each column for p values are ambiguous.

We added representative images of western blots to Figure 2DFigure.

We state in the text that eIF3A was not included in Figure 2 because we couldn't detect AHA-labelled eIF3A for technical reasons. Nonetheless, eIF3A did show significant increases in total protein after VE compared to control.

There are no lines to indicate which samples are compared because each data point is the ratio of NSP expression (B) (or total CPP expression (C)) in VE samples relative to a paired control. The band intensity of each NSP candidate protein was first normalized to its BSA loading control band (which was obtained after stripping the same membrane) and then that value was normalized to the β-tubulin loading control band from input/total protein samples. For samples in panel C, the band intensity of each candidate protein was normalized to its β-tubulin loading control band. For comparison between VE and control, we calculated the ratios of normalized VE and control intensities in each set of experiments.

We used t-tests or Mann-Whitney for statistical analysis because the comparisons between control and VE treatment for each candidate protein were made from paired control and experimental samples run in parallel for each data point. This information is now included the Figure legend and Materials and methods section.

5) For the proteomic screen, it should be explicitly stated how many animals were used and how many repeats were done in the main text or in the Figure legends. This information is suggested in Table 1 and can be found by reading the Materials and methods section very carefully but it is too important to be difficult to figure out.

We have added this information to the legend for Figure 1.

6) Figure 6 has no error bars or statistical analysis. Were multiple cells imaged per animal? Again, what is the n? ANOVA should be used instead of a t-test. The way the data is presented in the Figure legend of 7B,C is appropriate and all the legends in the manuscript should include this type of information.

In Figure 6, we present the proportion of all cells analyzed in each experimental condition that showed visual experience-dependent dendritic arbor growth. This type of data presentation does not include error bars. We added the following statistical analyses of these data. We analyzed the distribution using the Chi-square test of independence. The overall comparison yields a chi-square value of 28.98 and *p* =2.34×10^−5^, which is highly significant and indicates there is a significant difference between groups. We then examined which groups have significantly different proportions of the cells that respond to VE compared to the rest of the groups and determined p values for each comparison. In total there are 6 comparisons: Control MO vs. all others (p=0.0000026), eIF3A MO vs. all others (p=0.3772412), FUS MO vs. all others (p=0.8407727), RPS17 MO vs. all others (p=0.9773516), eIF3A MO+RPS17 MO vs. all others (p=0.2501854), and eIF3A MO+FUS MO+RPS17 MO vs. all others (p=0.0000791). *P* value has to be less than 0.05/6=0.0083 to be significant at the *P*<0.05 level with the Bonferroni correction. Both control morpholino and triple knockdown conditions have significantly different proportions of cells that respond to VE compared to the rest of the groups. We include this information in the text (subsection “Blocking both protein translation and RNA splicing has profound effects on visual experience-dependent structural and behavioral plasticity”), Figure legend and Figure 6.

We have modified the legends. The data and statistical analyses for all graphs are included the Figure source data files.

Single neurons were imaged per animal for all imaging experiments, as now stated in the Materials and methods section.

7) In Subsection “Blocking both protein translation and RNA splicing has profound effects on visual experience-dependent structural and behavioral plasticity” please clarify how dendritic growth rate is measured.

Dendritic growth rate is the change in total dendritic branch length (TDBL) over the 4h imaging interval, now stated in subsection “Reduced synthesis of RNA splicing and protein translation machinery blocks visual experience-dependent structural plasticity”. We also changed the y axis labels in Figure 4, Figure 5 and Figure 6 so that we uniformly use ‘Growth Rate (µm/4h)”.

8) There are confusing statements about Fus alternative splicing without any explanation. Why was the splice junction targeted, what are the splice variants, what splice is remaining here? This should be explained.

The FUS MO we used in this study is the same as the fusMO4 previously described (Dichmann and Harland, 2012). fusMO4 targets the first splice donor site of *fus*mRNA and is predicted to cause inclusion of intron 1 and reduce the amount of both *fus-a* and *fus-b* splice variants by premature termination (Dichmann and Harland, 2012). To evaluate MO efficiency, we used RT–PCR with primers covering the first four exons. We added this information to the Results section and the Materials and methods section.

9) While the observation of changes in growth were only down in the electroporated cells, changes in behavior presumably require electroporation of a large percentage of the tectal neurons. This should be discussed or explained.

We refer to our previous work demonstrating widespread delivery of morpholinos by electroporation and that this allows investigation of circuit function and behavior (subsection “Blocking both protein translation and RNA splicing has profound effects on visual experience-dependent structural and behavioral plasticity”).

10). Based on the time points that the authors have used, only a subset of proteins that are likely to be important for translation-dependent plasticity have probably been identified and this would miss many proteins that are synthesized quickly and then degraded, such as Arc and BDNF. Indeed, this same group examined a subset of selected proteins for increases in AHA-labeling acutely after learning (30 minutes of training, 2 hours afterwards), somewhat shorter than the time course examined here (4 hours afterwards). Relatively strong increases in CAMKII, GAD65, MEK1 and HDAC were observed. These were not replicated in the present study. Indeed, they still use CAMKII as their positive control in Figure 2 despite the lack of a change in their proteomics. Discussion of this is warranted.

We now discuss caveats mentioned in this and the following 2 comments in the Discussion section.

11) There is no evidence that there is a specific increase in the translation rate of the proteins identified, as opposed to an increase in the amount of protein made due to increased transcription followed by translation. An increase in transcription and basal translation would still lead to an increase in AHA-labeling, particular in a 4 hour labelling period. Again, this needs to be discussed.

We discuss this point in subsection “Proteomic Analysis of NSP Dynamics”.

12) Knockdowns of eIF3A and Rps17 are likely to decrease overall protein synthesis and this is a confound to the interpretation of these experiments. Indeed, the authors acknowledge this (haploinsufficiency of Rps17 causes small Drosophila; large effects on translation with knockdowns of eIF3A). However, whether or not the morpholinos cause a change in overall translation is not measured by the applicants (i.e. AHA labeling with immunocytochemistry comparing knockdown cells to adjoining cells without the knockdown). Since the plasticity requires protein synthesis, it is not clear that these experiments are not just the equivalent of adding anisomycin. A more general problem is that knocking down overall synthesis for a long period of time is not equivalent to knocking down the small increase in newly synthesized protein, since much larger changes in the proteome are induced by the manipulation than occur during memory. Both of these caveats should be discussed.

We added data assessing overall protein synthesis in the optic tectum using fluorescent non-canonical amino acid tagging (FUNCAT) to visualize *in vivo* AHA-labeled NSPs in the optic tectum (subsection “Blocking both protein translation and RNA splicing has profound effects on visual experience-dependent structural and behavioral plasticity” and Figure 6—figure supplement 1). These experiments show that eIF3A and RPS17 knockdown individually or in combination do not result in overall decreases in protein synthesis, suggesting that the effects of eIF3A and RPS17 knockdown in visual experience-dependent structural plasticity are not due to large scale decreases in protein synthesis. We discuss both of the caveats mentioned in subsection “Proteomic Analysis of NSP Dynamics”.

13) The methods in immunoblotting state that "Outliers, defined as those with AI greater that two SDs from the mean, were excluded from the analysis of total proteins". I assumed that many animals were pooled for these experiments (30-50 tecta), so I do not understand this statement.

This was an error to say ‘AI’, now corrected (subsection “Western Blot and Immunocytochemistry”): “Outliers, defined as those with normalized intensity (VE/control) greater that two SD from the mean, were excluded from the analysis of total proteins.”

14) Figure 2—figure supplement 1 is never mentioned in the text.

We mention this figure as figure supplement for Figure 2.

15) It is not clear what statistical test was used to sow that the effect of the double and triple Kos were larger than the single Kos as 7 there is no statistical comparison between the CMO and triple knockout. What was the definition of a 'non-plastic' cell. In Figure 7, the p value for the 'non-significant' result needs to be given as it appears to be close to significance. This should be given for all cells, or at least a general statement such as non-significant, p>0.5. P values less than or close to 0.1 are of borderline significance and in these cases, the actual p values should be given.

We mention the statistical tests in the Figure legends. The p values are provided in the Figure legends, in the Source data tables for each figure and in a separate table of statistics. Significance criterion is p<0.05.

We changed Figure 6 to be more clear.